# Intra-Trajectory Consistency for Reward Modeling

## Abstract

Reward models are critical for improving large language models (LLMs), particularly in reinforcement learning from human feedback (RLHF) and inference-time verification. Due to the prohibitive cost of fine-grained annotations, current reward models typically learn from holistic response scores to determine outcome rewards. However, this coarse-grained supervision makes it difficult for the reward model to identify which specific components within a response trajectory truly correlate with the final score, leading to poor generalization of unseen responses. In this paper, we introduce an intra-trajectory consistency regularization to propagate coarse, response-level supervision into fine-grained learning signals. Inspired by a Bayesian framework, our method enforces a simple principle: The rewards of adjacent generation processes should be more consistent when the connecting token has a higher generation probability. We apply the proposed regularization to the advanced outcome reward model, improving its performance on RewardBench. Furthermore, we demonstrate that the reward model trained with the proposed regularization yields better DPO-aligned policies and achieves superior best-of-N inference-time verification results. Our implementation code is provided in the supplementary material.

## 1 Introduction

Reward models offer a quantitative measure of the quality of LLM responses based on human preferences or correctness, making them instrumental in improving LLM performance through RLHF (Ouyang et al., 2022; Dai et al., 2024; Ethayarajh et al., 2024; Liu et al., 2025) or inference-time verification (Zhang et al., 2024b; Setlur et al., 2025). In RLHF, reward models provide feedback signals that guide LLMs to generate desirable responses via reinforcement learning. In inference-time verification, they rank or filter outputs to ensure the selection of the most appropriate responses. The generalization of the reward model is therefore critical, as these applications depend on reliable predictions for unseen responses (Gao et al., 2023; Yang et al., 2024b).

To enhance the generalization of the reward model, extensive efforts have been made in the literature, including ensemble techniques (Coste et al., 2024; Rame et al., 2024), data augmentation (Shen et al., 2024; Liu et al., 2024a), direct correction of bias caused by length (Dubois et al., 2024; Chen et al., 2024a), and hidden-state regularization (Yang et al., 2024b; Chen et al., 2024b). Generally, these methods use holistic human evaluations of responses to learn the rewards of responses (Sun et al., 2025; Yang et al., 2024b; Liu et al., 2024a). Despite their success, these models remain limited by coarse-grained supervision of the response-level scores, which hinders their ability to capture dependencies between responses and the processes properly. This may lead to overfitting to spurious features (Yang et al., 2024b), such as response length, instead of properly leveraging label-relevant components in the response trajectory, resulting in poor generalization to unseen responses. To identify content that influences the score of overall response, some approaches propose learning with process-level labels (Lightman et al., 2023; Wang et al., 2024b) or token-level labels (Yoon et al., 2024). However, in many practical scenarios, obtaining such fine-grained annotations proves prohibitively expensive (Zhang et al., 2024a).

To address these challenges, we propose establishing reward consistency between processes within the response trajectory, enabling response-level supervisory signals to propagate across processes and thereby enrich reward learning. Specifically, we utilize generation probabilities, which measure

Figure 1: Illustration of our proposed framework. In our framework, the reward model learns outcome rewards via a standard reward loss. We supplement this with an intra-trajectory consistency regularization term. The regularization enforces stronger reward consistency between adjacent processes with higher next-token probabilities from the generator.

the likelihood of a generator producing subsequent sequences, to capture inter-process dependencies. Inspired by Bayesian decomposition, we establish the connection between these generation probabilities and reward consistency: When a generator assigns a higher probability to a sequence of tokens, the rewards for the corresponding generation steps are more likely to be consistent. Moreover, to prevent severe misjudgment of reward consistency caused by low generation probabilities, we focus on adjacent processes with minimal content variation. These process pairs often exhibit semantic continuity and, consequently, tend to have comparable rewards.

To this end, we introduce intra-trajectory consistency regularization for reward modeling, termed ICRM. As shown in Figure 1, our framework consists of two components: a frozen generator that provides generation probabilities, and a reward model trained to predict outcome rewards. To propagate response-level supervisory signals throughout the process trajectory, the reward model is regularized to produce more consistent rewards for adjacent processes with higher next-token generation probabilities, thereby improving generalization without process-level labels.

Finally, the main contributions can be summarized as follows:

- **Exploration.** We investigate the relationship between next-token generation probabilities and reward consistency, drawing inspiration from a Bayesian framework.

- **Method.** We propose a regularization method that enforces higher reward consistency between adjacent processes with higher next-token generation probabilities, thereby more effectively utilizing response-level supervisory signals for better generalization.

- **Experiments.** We conduct extensive experiments to demonstrate that the proposed regularization improves the performance of the reward model in three evaluation tasks: standard reward modeling benchmarks, RLHF, and inference-time verification.

## 2 BACKGROUNDS

Given an input prompt $x$, a standard language generator $\theta_g$, such as many current LLMs (Mesnard et al., 2024; Dubey et al., 2024), generates token sequences autoregressively, predicting each token $y_t$ conditioned on the preceding subsequence $y_{1:t-1}$ until reaching either a termination token or a maximum length constraint. This process yields a complete output sequence $y = (y_1, \ldots, y_n) = y_{1:n}$. To enhance LLMs, many studies explore reinforcement learning training (Bai et al., 2022; Shao et al., 2024) or employ inference-time verification (Zhang et al., 2024b; Setlur et al., 2025). Both approaches require evaluating generated sequences, either through scoring or correctness assessment. This assessment is often referred to as the reward.

**Reward.** For an input $x$ with corresponding generated response sequence $y$ or a process $y_{1:k}$ consisting of the first $k$ tokens in $y$, reward functions can be categorized into two fundamental types:

outcome reward $r(x, y)$ and process reward $r(x, y_{1:k})$. The outcome reward evaluates the complete response based on its final solution quality (Uesato et al., 2022; Zhang et al., 2024b). In contrast, the process reward evaluates the scores of the intermediate processes within a response. Since it is not clear how to divide the processes of a common scenario, the response segment $y_{1:m}$ is considered valid. We should also note that in our definition, a full response can also be treated as a process. While this inclusive definition admits partial sentence fragments as independent processes, recent work in both RLHF (Zeng et al., 2024; Cui et al., 2025) and inference-time verification (Xu et al., 2025) has demonstrated the empirical effectiveness of such fine-grained reward signals.

**Reward Modeling.** Many existing methods train reward models $\theta_r$ using overall response-level annotations. The current dominant approach for reward modeling is the Bradley-Terry model (Sun et al., 2025). For this model, the training dataset $D_{tr}$ whose unit is a triple $(x, y^w, y^l)$, where $x$ represents an input or a prompt, $y^w$ is a chosen response for $x$, and $y^l$ is a rejected response for $x$. To distinguish between the chosen response and the rejected response for a given input, we can optimize the Bradley-Terry reward model with the objective

$$\mathcal{L}_{bt} = \mathbb{E}_{(x, y^w, y^l) \sim D_{tr}} \left[ -\log \sigma(\theta_r(x, y^w) - \theta_r(x, y^l)) \right], \quad (1)$$

where $\sigma$ is the sigmoid function. After optimization, the reward model can be used to provide outcome rewards for RLHF or inference-time verification. We discuss more specific reward model modeling methods, such as PRM (Lightman et al., 2023), in the Appendix A.

## 3 METHOD

This section introduces intra-trajectory consistency regularization to constrain intermediate generation processes that lack explicit labels. The method works by propagating response-level supervisory signals, leveraging the inherent reward consistency between steps in a generation trajectory. We first discuss the link between this reward consistency and generation probability (Section 3.1), then detail how it is implemented to regularize the reward model (Section 3.2), and finally integrate the proposed regularization into our training framework to learn more reliable rewards (Section 3.3).

### 3.1 ESTABLISHMENT OF REWARD CONSISTENCY

Traditional reward modeling uses coarse response-level scores (e.g., pairwise preferences) (Sun et al., 2025; Yang et al., 2024b; Liu et al., 2024a), making it difficult to assess fine-grained correctness (Wu et al., 2023). To introduce fine-grained signals, we propose establishing reward consistency relations between processes with the same response trajectory. This framework enables response-level supervisory signals to propagate throughout the trajectory, providing additional signals for reward learning. These derived signals help the model better capture contextual dependencies between processes. Besides, this is achieved without requiring additional manual annotation.

To establish the intra-trajectory consistency, we propose to leverage generation probabilities, the likelihood of a generator producing each subsequent sequence, to reflect the reward dependencies between processes. For example, research in the safety domain (Qi et al., 2025) demonstrates that certain intermediate processes, such as phrases like "Sure, here is a detailed guide," often precede hazardous completions in response to harmful queries. Therefore, linking reward relationships between processes with their underlying generation probabilities is possible.

To achieve this objective, we formalize our approach by making key assumptions about the generation of the response. Specifically, we assume that responses are generated by a generator $\theta_g$ with an estimable conditional probability distribution, meaning each new token depends probabilistically on all previously generated tokens. This assumption aligns with the autoregressive nature of modern LLMs. Under these conditions, we can connect process $y_{1:m}$ and its subsequent process $y_{1:n}$ (where $m < n$) for input $x$ with the generation probability through Bayesian decomposition:

$$P(e|x, y_{1:m}) = P(e|x, y_{1:n})P(x, y_{1:n}|x, y_{1:m}) + \sum_{\bar{y}_{1:n} \in \bar{Y}_{1:n}} P(e|x, \bar{y}_{1:n})P(x, \bar{y}_{1:n}|x, y_{1:m}), \quad (2)$$

where $P(e|x, y)$ denotes the conditional probability of any event $e$ occurring given $(x, y_{1:m})$. $P(x, y_{1:n} \mid x, y_{1:m})$ represents the generation probability of sequence $y_{1:n}$ conditioned on $(x, y_{1:m})$,

as computed by the generator. Since $y_{1:n}$ is the successor of $y_{1:m}$, $P(x, y_{1:n} \mid x, y_{1:m})$ also equals to $P(y_{m:n} \mid x, y_{1:m})$. $\bar{Y}_{1:n}$ denotes the set of all possible sequences with length $n$ excluding $y_{1:n}$.

Eq. 2 formalizes the connection between a process and its future outcomes. The intuition is that the process reward $r(x, y_{1:m})$ should reflect its potential to evolve into a preferred final response. This mirrors Q-values (Wang et al., 2024b; Setlur et al., 2025), which estimate the expected return from a given state. We therefore adopt the Q-value analogy from (Li & Li, 2025) and model the process reward as the likelihood of ultimately generating a preferred response from the current process $y_{1:m}$. While a scalar reward is not a true probability, enforcing this consistency provides a well-founded mechanism for propagating coarse, response-level supervision to the process level. Thus, we acquire our framework with two informal assumptions: first, that the output of reward model can be treated as a probabilistic score, and second, that the generation probabilities of subsequent tokens are accessible from a generator model. Then letting event $e$ denote the generation of preferred response, we can replace $P(e|x, y_{1:m})$ with $r(x, y_{1:m})$, represented as:

$$r(x, y_{1:m}) = r(x, y_{1:n})P(x, y_{1:n}|x, y_{1:m}) + \sum_{\bar{y}_{1:n} \in \bar{Y}_{1:n}} r(x, \bar{y}_{1:n})P(x, \bar{y}_{1:n}|x, y_{1:m}). \quad (3)$$

From Eq. 3, since $\sum_{\bar{y}_{1:n} \in \bar{Y}_{1:n}} P(\bar{y}_{1:n}|x, y_{1:m}) + P(y_{1:n}|x, y_{1:m}) = 1$, as the generation probability $P(x, y_{1:n}|x, y_{1:m})$ increases, the contribution of alternative completions $\bar{y}_{1:n}$ to the reward $r(x, y_{1:m})$ diminishes. Therefore, the reward $r(x, y_{1:m})$ becomes increasingly dominated by $r(x, y_{1:n})$, reducing the variance of the reward and leading to higher consistency between $r(x, y_{1:m})$ and $r(x, y_{1:n})$. Thus, generation probability and reward consistency can be linked. Compared with direct learning of Q-value, Eq. 3 allows the use of a generator's probabilities directly for regularization rather than relying on full rollouts and labeling.

The above analysis also implies an issue: When the generation probabilities between processes are low, reward similarity estimation may be unreliable. To address this, we incorporate reward consistency between semantically related processes. Inspired by text augmentation methods (Qu et al., 2021; Bayer et al., 2022), where partial masking preserves semantics, we assume that process semantics remain stable under limited suffix additions. This semantic invariance suggests correlated rewards between successive processes ($y_{1:m-1}$ and $y_{1:m}$). Consequently, we focus on generation probabilities and reward consistency for adjacent processes as a more tractable approach. The impact of the token distance is discussed in Appendix D.8.

## 3.2 INTRA-TRAJECTORY CONSISTENCY REGULARIZATION

Building on the above analysis, we propose *intra-trajectory consistency regularization* to enforce more consistent rewards between adjacent processes with higher next-token generation probabilities. We subsequently present our reward formulation of processes under the Bradley-Terry framework, followed by the corresponding optimization objective.

For reward formulation, we note that standard sigmoid reward outputs under the Bradley-Terry framework often saturate near the boundary values, i.e., 0 or 1. Thus, to ensure our regularization focuses on learning meaningful relative differences between adjacent process rewards rather than pushing absolute scores to their limits, we introduce a mean-centered calibration technique. This technique uses the average process reward from the opposing trajectory (e.g., the rejected response) as a dynamic, data-dependent baseline to calibrate the process rewards of the current trajectory (e.g., the chosen response), and vice versa. Based on this calibration, the process rewards will have distinctiveness. This mutual calibration encourages the model to learn a more well-separated reward space. For a process $y_{1:m}^w$ in the chosen response $y^w$, we define its calibrated reward as:

$$\hat{r}(x, y_{1:m}^w) = \sigma(\theta_r(x, y_{1:m}^w) - \frac{1}{|y^l|}\sum_{k=1}^{|y^l|} \theta_r(x, y_{1:k}^l)), \quad (4)$$

where $|y|$ denotes the length of sequence $y$. The mean value serves only as a calibration term and is excluded from gradient computation. Analogously, for a process $y_{1:m}^l$ in the rejected response $y^l$,

the calibrated reward is:

$$\hat{r}(x, y_{1:m}^l) = \sigma(\theta_r(x, y_{1:m}^l) - \frac{1}{|y^w|}\sum_{k=1}^{|y^w|}\theta_r(x, y_{1:k}^w)). \tag{5}$$

Building upon the calibrated rewards of processes, we can introduce a method to enforce reward consistency between adjacent processes. A direct method is to minimize calibrated reward distances (e.g., absolute differences) between adjacent processes. However, this method is ineffective under stochastic rewards (e.g., randomly initialized values), as forcing consistency between arbitrary rewards has limited meaning. Inspired by (Sohn et al., 2020; Zhang et al., 2021), we address this limitation through a mutually weighted binary cross-entropy loss that both learns semantically meaningful process rewards and promotes the reward consistency between adjacent processes.

Specifically, for a triple preference $(x, y^w, y^l)$, we assign process-level pseudo-labels identical to the response label $s$, where $s = 1$ indicates a chosen response $y^w$ and $s = 0$ indicates a rejected one $y^l$. The weighting mechanism for a pair of adjacent processes $(y_{1:k-1}, y_{1:k})$ in a response combines two factors: (1) the probability of the next token $P(x, y_{1:k}|x, y_{1:k-1}) = P(y_k|x, y_{1:k-1}) = \theta_g(y_k \mid x, y_{1:k-1})$ from the generator $\theta_g$, and (2) the prediction confidence of the calibrated reward of another paired process. Formally, for $y_{1:k-1}$ in the pair, its weight is computed as:

$$w(k \to k-1, s) = \theta_g(y_k|x, y_{1:k-1}) \cdot (s \cdot \hat{r}(x, y_{1:k}) + (1-s) \cdot (1 - \hat{r}(x, y_{1:k}))). \tag{6}$$

Similarly, for $y_{1:k}$ in the pair, its weight is formulated as:

$$w(k-1 \to k, s) = \theta_g(y_k|x, y_{1:k-1}) \cdot (s \cdot \hat{r}(x, y_{1:k-1}) + (1-s) \cdot (1 - \hat{r}(x, y_{1:k-1}))). \tag{7}$$

These weights are not used for gradient computation. When $s = 1$, $y = y^w$. When $s = 0$, $y = y^l$. Finally, let $\breve{r}(\cdot) = 1 - \hat{r}(\cdot)$, the regularization loss for all training triples $(x, y^w, y^l)$ is:

$$\mathcal{L}_{reg} = \mathbb{E}_{(x,y^w,y^l)\sim D_{tr}}[-\sum_{k=2}^{|y^w|}w(k \to k-1, 1)\log\hat{r}(x, y_{1:k-1}^w) + w(k-1 \to k, 1)\log\hat{r}(x, y_{1:k}^w)$$

$$-\sum_{k=2}^{|y^l|}w(k \to k-1, 0)\log\breve{r}(x, y_{1:k-1}^l) + w(k-1 \to k, 0)\log\breve{r}(x, y_{1:k}^l)]. \tag{8}$$

In Eq. 8, the binary classification loss deviates the random process rewards to meaningful ones. Besides, since the losses of adjacent processes are mutually weighted by rewards from each other, their rewards can gradually become similar. The degree of this consistency constraint is implicitly governed by their next-token generation probabilities. Consequently, Eq. 8 prioritizes meaningful reward consistency for adjacent processes with higher next-token generation probabilities.

### 3.3 OVERALL TRAINING FRAMEWORK

This section details how we enhance the standard Bradley-Terry reward model with an intra-trajectory consistency regularizer to learn more robust outcome rewards. Since existing reward modeling datasets are often aggregated from diverse sources, including some unavailable models, we begin by performing supervised fine-tuning (SFT) on a pre-trained language model to derive a generator $\theta_g$. This generator is optimized to align with the training dataset's generation probability distribution. Then this fine-tuned generator can provide next-token generation probability for computing $L_{reg}$. In cases where the data source is a single known model, we use it directly as the generator (see Appendix D.7 for a discussion on generator mismatch). After acquiring the generator, we use two objectives to train the reward model: (1) a reward modeling loss $L_{bt}$ applied to the entire response, which facilitates learning the outcome rewards; and (2) a regularization $L_{reg}$ applied across processes. Finally, the overall loss to update the model is computed as:

$$\mathcal{L}_{toal} = (1-\alpha)\mathcal{L}_{bt} + \alpha\mathcal{L}_{reg}, \tag{9}$$

where $\alpha$ is a balance hyper-parameter. Compared to solely optimizing $\mathcal{L}_{bt}$, optimizing Eq. 9 enables the model to focus on finer-grained signals. After training, the model can be used to provide outcome rewards. As an extension, we provide the discussion on computational cost and possible efficiency improvements in Appendix C.4.

Table 1: Accuracy results on RewardBench with 40K training samples from Unified-Feedback. The base model is Gemma-2B-it. The best results in a column of a series are highlighted in bold. * indicates that the result is copied from (Yang et al., 2024b).

| Reward Model | Chat | Chat-Hard | Safety | Reasoning | Average |
|---|---|---|---|---|---|
| Classifier + margin* | **97.2** | 37.5 | 56.8 | 72.7 | 66.1 |
| Classifier + label smooth* | 91.6 | 39.0 | 53.8 | 60.2 | 61.1 |
| Classifier + Ensemble* | 96.1 | 38.2 | 58.8 | 67.6 | 65.2 |
| GRM* | 94.7 | 40.8 | 65.4 | **77.0** | 69.5 |
| GRM (reproduced) | 96.8±0.1 | 41.1±0.2 | 80.6±0.2 | 73.9±0.2 | 73.1±0.1 |
| ICRM (Ours) | 95.0±0.1 | **48.1**±0.9 | **84.3**±0.2 | 75.6±0.3 | **75.8**±0.3 |

Table 2: Accuracy results on RewardBench with 400K training samples from Unified-Feedback. The base model is Gemma-2B-it. The best results in a column of a series are highlighted in bold. * indicates that the result is copied from (Yang et al., 2024b).

| Reward Model | Chat | Chat-Hard | Safety | Reasoning | Average |
|---|---|---|---|---|---|
| Classifier + margin* | 89.7 | 47.1 | 70.7 | 43.6 | 62.8 |
| Classifier + label smooth* | 94.1 | 47.1 | 67.5 | 79.7 | 72.1 |
| Classifier + Ensemble* | 89.6 | **50.2** | 72.7 | 59.0 | 69.3 |
| GRM* | **96.1** | 40.1 | 80.3 | 69.3 | 71.5 |
| GRM (reproduced) | 95.3 | 43.2 | 78.9 | 75.2 | 73.2 |
| ICRM (Ours) | 95.5 | 44.5 | **84.5** | **78.2** | **75.7** |

## 4 EXPERIMENTS

### 4.1 EXPERIMENTAL SETUP

**Datasets.** (1) We train the reward models on the widely-used Unified-Feedback[1] and Skywork[2]. The data of these two datasets are derived from multiple LLMs. Thus, to verify the proposed method in the situation where the generation distribution is known, we adopt Qwen2.5-7B-Instruct (Yang et al., 2024a) to generate 4 responses for each question in the training set of the prm-800k dataset (Lightman et al., 2023) and label these responses with gold answers. These generated responses constitute the Qwen-Generated dataset. (2) For experiments of RLHF, we sample about 500 prompts from the prompts of RewardBench as the test set and the rest as the training set. (3) For experiments of inference-time verification, we evaluate the reward model on MATH-500 (Hendrycks et al., 2021), with BON datasets from (Yuan et al., 2024), containing processes generated by both Mistral-7B-Instructor-v0.2 (Jiang et al., 2023) and Llama-3-8B-Instruct (Dubey et al., 2024).

**Training Details.** (1) We train the proposed reward modeling method based on GRM (Yang et al., 2024b), which achieves competitive results in the RewardBench benchmark. The resulting reward model is termed **ICRM**. We validate the proposed method on Gemma-2B-it (Mesnard et al., 2024), Llama3-8B-instruct (Dubey et al., 2024), and Qwen-1.5B-Instruct. The hyperparameter $\alpha$ is set to 0.1 in all of our experiments, with analysis shown in Appendix D.6. Additional training configurations are detailed in Appendix C. (2) For RLHF, we adopt DPO strategy (Rafailov et al., 2023) to train the policy model with Gemma-2B-it as the initial model. Following (Dong et al., 2024; Liu et al., 2024a), we generate 8 responses for each prompt in the training set of RLHF. Then these responses are scored by the 2B reward model trained with Unified-Feedback 400K. The best-worst response pairs for each prompt are used to train the DPO policy. All experiments are implemented on at most two A800 GPUs, each with 80GB of memory.

**Baselines and Evaluation Details.** In this paper, we evaluate the proposed method in three tasks: the standard RewardBench benchmark (Lambert et al., 2024), RLHF, and inference-time verifica-

---

[1]https://huggingface.co/datasets/llm-blender/Unified-Feedback
[2]https://huggingface.co/datasets/Skywork/Skywork-Reward-Preference-80K-v0.2

Table 3: Accuracy results on RewardBench with Llama3-8B-instruct. Best results are highlighted in bold. "avg" refers to the use of an exponential moving average (EMA) of the rewards from the trailing tokens during inference, with the smoothing applied backward from the last token and a decay factor of 0.5.

| Reward Model | Chat | Chat-Hard | Safety | Reasoning | Average |
|---|---|---|---|---|---|
| SyncPL-o1 Liang et al. (2025) | 93.9 | 73.2 | 85.8 | 83.7 | 84.2 |
| PRISM Ye et al. (2025) | **98.7** | 68.3 | **91.1** | 93.1 | 87.8 |
| GRM | 95.5 | 74.1 | 86.6 | 89.0 | 86.3 |
| GRM-avg | 96.9 | 74.1 | 85.0 | 91.2 | 86.8 |
| ICRM (Ours) | 95.2 | 75.9 | 86.2 | 89.7 | 86.8 |
| ICRM-avg | 96.1 | **78.1** | 87.3 | **95.0** | **89.1** |

tion. Evaluation of more reward benchmarks is provided in Appendix D.13. (1) For the Reward-Bench benchmark, we consider Classifier trained with Eq. 1, Classifier+Margin (Touvron et al., 2023), Classifier+Label Smooth (Wang et al., 2024a), Classifier+Ensemble (Coste et al., 2023), GRM (Yang et al., 2024b), SyncPL-o1 (Liang et al., 2025), and PRISM (Ye et al., 2025) as baselines. (2) For RLHF, inspired by (Dubois et al., 2023; Yang et al., 2024b), to avoid the high costs of human evaluation, we employ the QRM-Llama3.1-8B model [3] as a gold scoring model according to its strong performance on the RewardBench benchmark. The trained DPO model generates responses for the prompt in the test set by greedy sampling. (3) When evaluating inference-time verification, we adopt the best-of-N (BON) metric, which measures the probability that the response selected by the reward model from N alternative responses is the correct answer or the best solution.

## 4.2 EVALUATION ON REWARD MODELING BENCHMARK

**Results on ReardBench Benchmark.** As shown in Tables 1, 2, and 3, our method achieves higher average scores on the RewardBench benchmark than all baselines under identical settings. This improvement is statistically significant (p value=0.002, Table 1), indicating that our proposed regularization enhances the reward model's generalization. Given these significant results and the high cost of training LLM, we do not conduct further experiments on a larger scale.

**Comparison of Different Sizes of Training Samples.** Following (Yang et al., 2024b), we also compare the performance of the proposed method under different sizes of training samples. In Table 1 and Table 2, we present results for both 40K and 400K training samples. Additional results for other training sizes are provided in Appendix D.1. These results demonstrate that our method consistently achieves a higher average score than the GRM baseline, demonstrating its effectiveness across different data scales. See Appendix D.1 for more results.

**Utilization of the Process Rewards.**
Evaluating response correctness with process rewards is well-established in reasoning tasks (Lightman et al., 2023; Li & Li, 2025). Since our method also utilizes process rewards, we integrate them to examine their impact. We compute an exponentially weighted moving average of rewards, starting from the final token of each response and proceeding backward through the sequence. Given a response with length $m$ and the average decay $d$, the

Table 4: Results of DPO policy with guidance from different reward models. "Win ratio", "Tie ratio", and "Lose ratio" represent the proportions of comparisons in which a model's outputs are preferred (win), deemed equivalent (tie), or dispreferred (lose) relative to another model's outputs, respectively.

| Reward Model | Win ratio↑ | Tie ratio | Lose ratio↓ |
|---|---|---|---|
| GRM | 47.3 | 2.1 | 50.6 |
| ICRM (Ours) | 50.6 | 2.1 | 47.3 |

average reward is $r = \sum_{i=1}^{m} d^{m-i} r(x, y_{1:i})$. The results are shown in Table 3. Our findings reveal that incorporating this strategy leads to improvements for both GRM and our method in the reasoning group, with occasional gains in other groups. Crucially, our method surpasses GRM in average when using this approach. These results suggest that our approach learns reliable process rewards to some extent. More analysis is provided in Appendix D.9 and Appendix D.10.

---

[3] https://huggingface.co/nicolinho/QRM-Llama3.1-8B-v2

Table 5: Best-of-N (BON) inference-time verification results of responses from different polices and pass@N. All reward models are implemented using Qwen-1.5B-Instruct as the base model and trained on the Qwen-Generated dataset.

| Policy | Reward Model | Pass@2 | Pass@4 | Pass@8 | Pass@16 | Average |
|---|---|---|---|---|---|---|
| Mistral-7B-Instruct-v0.2 | GRM | 11.8 | 11.8 | 12.6 | 14.2 | 12.6 |
| | ICRM (Ours) | 11.8 | 12.8 | 14.6 | 14.0 | 13.3 |
| Llama-3-8B-Instruct | GRM | 45.6 | 46.8 | 49.2 | 45.0 | 46.7 |
| | ICRM (Ours) | 45.8 | 47.2 | 51.2 | 50.0 | 48.6 |

Table 6: Ablation study for the proposed regularization. "w/o adjacent reg" means that the reward of another process is not used for weighting in the proposed regularization. "w/o generation reg" means that generation probability is not used in the proposed regularization. "L1 loss" means using L1 loss to directly align rewards of adjacent processes. Training dataset is 40K samples from Unified-Feedback, and model is Gemma-2B-it.

| Method | Chat | Chat-Hard | Safety | Reasoning | Average |
|---|---|---|---|---|---|
| w/o adjacent reg | 96.1 | 43.2 | 80.8 | 74.0 | 73.5 |
| w/o generation reg | 95.2 | 46.9 | 83.5 | 75.2 | 75.2 |
| L1 loss | 96.4 | 42.1 | 80.3 | 74.3 | 73.3 |
| Overall | 95.0 | 48.2 | 84.2 | 75.8 | 75.8 |

### 4.3 EVALUATION ON RLHF

We also analyze the performance of policies optimized under different reward models, with results presented in Table 4. Our analysis reveals two findings: (1) the policy induced by the proposed reward model achieves higher gold scores compared to the baseline GRM, and (2) demonstrates superior prompt-conditional response quality, generating higher-scoring outputs for identical prompts. These results collectively validate the enhanced capability of our reward model in RLHF pipelines, particularly in its ability to guide policy to generate more desirable responses. We also provide RLHF results evaluated by humans in Appendix D.11.

### 4.4 EVALUATION ON INFERENCE-TIME VERIFICATION

We assess our reward model's capacity for inference-time verification using Best-of-N (BoN) sampling. The results in Table 5 show our method consistently outperforming the GRM baseline in mean accuracy on both Mistral-Instructor-v0.3 and Llama-3-8B-Instruct. Notably, the generator for this reward model was the same LLM that produced its training data, validating our method's ability to leverage accessible data generators for improved performance. We provide additional BoN results in Appendix D.4 and a comparison to process-based reward models in Appendix D.5.

### 4.5 EMPIRICAL ANALYSIS

**Ablation Study of Main Components.** The proposed intra-trajectory consistency regularization loss incorporates two key weighting components in Eq. 6 and Eq. 7 : (1) weights derived from rewards of adjacent processes, and (2) weights based on generation probabilities. To evaluate their respective contributions, we conduct ablation studies in Table 6, where each component is removed individually. The results demonstrate that removing either weighting component typically degrades performance, with the weights from rewards of adjacent processes showing a more substantial impact. These findings underscore the importance of both weighting mechanisms in the proposed intra-trajectory reward consistency regularization term.

**Evaluation with L1 Reward Alignment** To validate the effectiveness of our proposed regularization, we compare it against a standard L1 loss baseline. While L1 loss enforces consistency between adjacent process rewards, it imposes a uniform constraint that treats all tokens identically. This rigid-

Table 7: Ablation study on the effect of the mean-centered calibration technique. This model is trained on 40k Unified-Feedback samples using a Gemma-2B backbone under identical settings.

| Method | Accuracy |
|---|---|
| ICRM (Ours) | 75.8 |
| GRM (baseline) | 73.1 |
| GRM with mean-centered calibration | 44.2 |

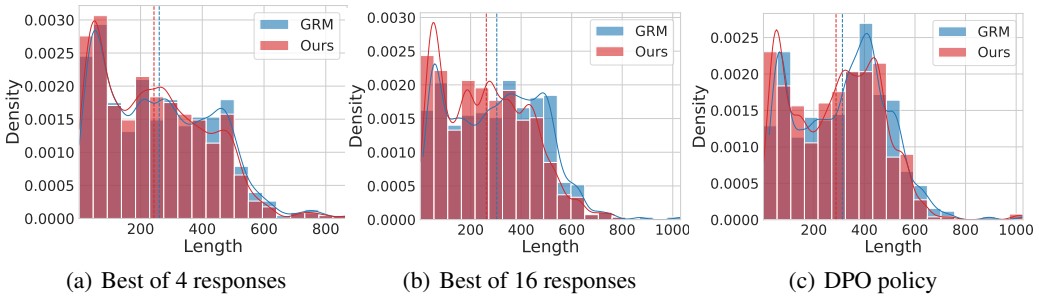

(a) Best of 4 responses  (b) Best of 16 responses  (c) DPO policy

Figure 2: Length distribution of responses of various policies induced by GRM and ICRM. Average length is marked by the dashed line. Prompts for generation are derived from the test set in the RLHF experiments. Generation model is Gemma-2b-it.

ity can differ from optimal alignment, potentially hindering performance. In contrast, our proposed method adopts a more adaptive approach via a mutually weighted learning objective. By scaling the regularization weight according to the next-token probabilities, we prioritize consistency constraints on tokens where the model exhibits higher confidence. As demonstrated in Table 6, this tailored strategy yields superior performance, achieving higher average accuracy than the L1 baseline.

Ablation on Mean-Centered Calibration To disentangle the effects of our proposed consistency regularization from the mean-centered calibration technique, we conduct an additional ablation study. Specifically, we create a variant of the baseline GRM by replacing the calibration term in its outcome-based objective (Eq. 1) with the mean-centered calibration used in our method (Eqs. 4 and 5). The results, presented in Table 7, show a drastic performance degradation for the modified GRM. This outcome substantiates our core argument. To elaborate, our mean-centered calibration term is, by definition, the average of all *process rewards* within a response. The magnitude of this average can naturally differ from the final *outcome reward* of that same response. Consequently, applying this term to an outcome-based objective like the Bradley-Terry loss creates a conceptual mismatch: it attempts to calibrate an outcome reward using a process-based average. This misalignment introduces noise and disrupts the optimization process. In contrast, our method applies this calibration within a regularization term that compares adjacent *process rewards*. This aligns with our regularization goals.

**Length Analysis.** Length represents a common superficial feature that reward models may exploit, often manifesting as a preference for longer responses (Shen et al., 2023). To investigate this length bias, we analyze response length distributions across different policy outputs (Figure 2). Our results demonstrate that policies optimized under our reward model consistently produce shorter responses than the baseline. This finding indicates that the proposed method has the potential to reduce the model's reliance on length as a proxy for response quality.

**Visualization of Process Rewards.** To elucidate the relationship between process-level rewards and response labeling learned by the proposed method, we visualize reward trajectories for two minimally contrasting responses that differ only in a few critical words (Figure 3). Our analysis reveals that: (1) the model systematically assigns higher rewards to processes containing semantically favorable words in context, and (2) for rejected responses, rewards exhibit gradual degradation rather than immediate drops. For instance, the token "ignore" triggers only a mild penalty, while subsequent unfriendly terms like "cruel" induce sharper declines. This demonstrates our method's capacity to develop nuanced and context-sensitive reward signals at the process level.

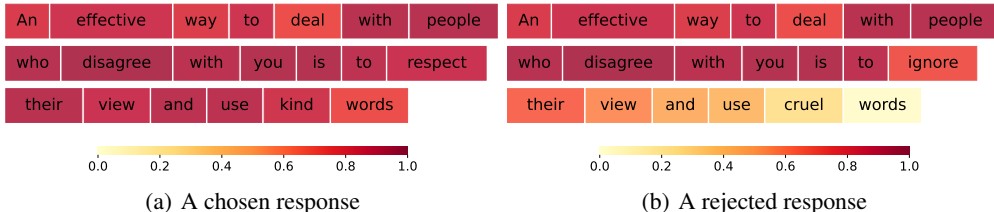

(a) A chosen response                    (b) A rejected response

Figure 3: Heatmap of the rewards acquired by ICRM for different processes, in which the reward of a process is shown in the last token of the process, and darker colors indicate higher rewards. The prompt of the two responses is "What is an effective way to deal with people who disagree with me?". The left response (with "respect" and "kind") is preferred as the chosen response over the right (with "ignore" and "cruel") due to its more positive wording concerning the context.

## 5 CONCLUSION

In this paper, we aim to address the limitation of conventional outcome reward modeling that fails to capture fine-grained details within a response with coarse, response-level supervision. We introduce a novel intra-trajectory consistency regularization to inject a finer-grained signal into the learning process. Motivated by the Bayesian framework, the proposed regularization enforces that adjacent processes with higher next-token generation probabilities maintain more consistent rewards. Experimental results on the RewardBench benchmark, RLHF, and inference-time verification validate the effectiveness of the proposed regularization in improving the advanced reward model. While our method's reliance on the generation probabilities from a generator introduces a computational overhead, this cost can be substantially mitigated through batch pre-processing. Our current validation is conducted on smaller-scale models due to resource constraints; therefore, scaling our method to larger models remains a promising direction for future work.

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

## A RELATED WORKS

To begin with, we discuss the related works to this study.

### A.1 GENERALIZATION OF REWARD MODELS.

The generalization of reward models to unseen responses is essential for improving their robustness in RLHF and inference-time verification (Gao et al., 2023; Yang et al., 2024b). To improve it, multiple approaches have been developed, such as ensemble techniques (Eisenstein et al., 2023; Rame et al., 2024), data augmentation (Shen et al., 2024; Liu et al., 2024a), direct correction of measurable bias (Dubois et al., 2024; Chen et al., 2024a), and hidden-state regularization (Yang et al., 2024b). For example, (Coste et al., 2024) proposes to learn multiple estimators and combine them to improve the robustness of the rewards. (Liu et al., 2024a) introduces a data augmentation approach derived from the causal framework to differentiate between contextual signals and context-free artifacts. (Chen et al., 2024a) proposes a framework trained to predict both rewards and lengths so that it can disentangle the representation of the content quality from the lengths of responses. (Yang et al., 2024b) introduces SFT and DPO losses to regularize the hidden states of reward models. While these methods strengthen reward models for aligning LLM, their effectiveness remains constrained by sparse response-level supervision, limiting further generalization. Besides, compared with GRM (Yang et al., 2024b), ICRM regularizes the final process rewards of the reward model. Consequently, the two methods may complement each other by operating at the feature level and the prediction level, respectively.

### A.2 OUTCOME REWARDS AND PROCESS REWARDS.

Outcome and process rewards represent two fundamental paradigms in reinforcement learning and AI alignment (Schulman et al., 2017; Im & Li, 2024; Uesato et al., 2022). Outcome rewards, which evaluate a task's final result, are straightforward to specify and require minimal annotation effort. However, the sparse signals they provide often fail to guide agents toward desirable behaviors in complex, long-horizon tasks (Lightman et al., 2023). In contrast, process rewards evaluate intermediate steps, thereby providing denser feedback. This approach is critical for autoregressive generation, where it can enforce properties like coherent reasoning by capturing causal dependencies between tokens (Li & Li, 2025).

Although process-based reward models are effective, they typically demand expensive, step-level annotations (Wang et al., 2024b; Luo et al., 2024). Efforts to automate this annotation, such as scoring intermediate steps based on the final outcome, are often suboptimal without a perfect oracle and computationally intensive, requiring numerous generated responses (Wang et al., 2024b). A different strategy involves using an external LLM to minimally revise a "rejected" response, from which token-level preference labels are algorithmically derived (Yoon et al., 2024). This method, however, risks inheriting biases from the revising LLM, especially in cases of ambiguous human preference. Given these limitations, merely learning from response-level annotation remains important. Therefore, the proposed method's ability to achieve the propagation of fine-grained process signals using only coarse, response-level labels becomes meaningful.

## B REPRODUCIBILITY STATEMENT

To ensure the reproducibility of our work, we provide a detailed account of our methodology and experimental setup. The proposed method is thoroughly described in Section 3. All experimental details—including dataset sources, base models, software frameworks, evaluation procedures, and hyperparameter settings—are specified in Section 4.1 and further elaborated in Appendix C. Furthermore, our complete implementation code is included in the supplementary material to facilitate the replication of our main findings.

## C IMPLEMENTATION DETAILS

In this section, we provide more implementation details.

Table 8: Common hyper-parameters in the experiments.

| | |
|---|---|
| Quantization for training | bf16 |
| LoRA r | 32 |
| LoRA alpha | 64 |
| Optimizer | Adamw |
| Learning rate | 1e-5 |
| Learning rate scheduler | cosine |
| Warmup ratio | 0.03 |

### C.1 IMPLEMENTATION FOR REWARD MODELING WITH UNPAIRED DATA.

When paired data are not available, discriminative reward modeling is usually used to learn the reward. The training dataset $D_{tr}$ consists of triples $\{x^i, y^i, c^i\}_{i=1}^N$, where $x^i$ represents input for $i^{th}$ example, $y^i$ is a response for $x^i$, and $c^i$ is a gold binary classification label for $x^i$. To estimate the quality of a response for an input, we can train a discriminative reward model $\theta_r$ with the binary cross-entropy loss, namely,

$$\mathcal{L}_d = \mathbb{E}_{(x,y,c)\sim D_{tr}}[-c \cdot \log(\sigma(\theta_r(x,y))) - (1-c) \cdot \log(1 - \sigma(\theta_r(x,y)))]. \qquad (10)$$

In this case, when computing the proposed regularization term, no calibration is required for the reward, i.e, $\hat{r}(x,y) = \sigma(\theta_r(x,y))$ in Eq. 8. Finally, based on discriminative reward modeling, we only need to add $\mathcal{L}_d$ and the proposed regularization term, similar to Eq. 9.

### C.2 TRAINING DETAILS OF REWARD MODELS

Our code is based on LlamaFactory (Zheng et al., 2024), a powerful code framework that supports multiple LLMs training strategies. The version number we adopted is 0.9.1.dev0. The LLMs involved in the experiments are all downloaded from HuggingFace. In the experiments, the models are trained with LoRA (Hu et al., 2022) for efficient fine-tuning. We employ the DeepSpeed framework[4] to enhance training efficiency and reduce GPU memory requirements. For models with fewer than 2B parameters, we utilize the ZeRO-2 offload configuration. When training includes 8B models, we adopt the ZeRO-3 offload setting.

Our reward model training procedure primarily follows GRM (Yang et al., 2024b), specifically adopting their GRM-sft configuration. GRM-sft incorporates an additional SFT loss alongside the standard reward loss to regularize hidden-state representations, with a default weighting factor of 0.001. The reward model is initialized using the backbone of a pre-trained LLM with an additional randomly initialized linear logistic regression layer (dropout=0.1). Common hyper-parameters are detailed in Table 8. Key variations include batch size and training epochs: for datasets less than 40K, we use batch size 12 with 2 epochs; other experiments employ batch size 24 with 1 epoch. Notably, the proposed method occasionally requires a generator during reward model training. The generator for the 2B model is trained with 40K samples from the Unified-Feedback dataset. The generator for the 8B model is trained with 40K samples from the Unified-Feedback dataset and the Skywork dataset. When training, it shares the same configuration as the reward model (Table 8), except for a reduced learning rate (1e-7) and a single epoch to prevent overfitting. For the reward model that is only trained with the Unified-Feedback dataset, we set the sequence cutoff length to 1,024 tokens. All other reward models and generators use a 2,048-token cutoff to handle long-form data better.

### C.3 HYPER-PARAMETERS FOR RLHF.

In RLHF experiments, the process consists of two key components: response generation and subsequent training using these generated responses. For data generation, we employ a temperature of 1.0 and top-p sampling (p=1.0), producing 8 responses per prompt with a maximum length of 1,536 tokens. These responses are then evaluated by our 2B reward model trained on the Unified-Feedback

---

[4]https://github.com/deepspeedai/DeepSpeed?tab=readme-ov-file

Table 9: Accuracy results on RewardBench with different sizes of training samples from the Unified-Feedback dataset. The base model is Gemma-2b-it.

| Reward Model | 4K | 10K | 40K | 400K | Average |
|---|---|---|---|---|---|
| GRM | 59.5 | 64.1 | 73.0 | 73.2 | 67.5 |
| ICRM (Ours) | 61.3 | 64.3 | 75.8 | 75.7 | 69.3 |

400K dataset. We select the best-worst response pairs from each prompt for DPO policy training. The DPO configuration largely follows the settings in Table 8, with two modifications: (1) a learning rate of 5e-6, and (2) the addition of a 0.1 scaling factor for comparing current and reference model probabilities. All DPO experiments run for 2 epochs.

### C.4 Discussion on Computational Cost

All experiments are implemented in at most two NVIDIA RTX A800 GPUs, which have about 160 GB of memory. During training of the 2B GRM reward model, the baseline in this paper, with a batch size of 12, we observe a training speed of 4.6 seconds per iteration. Under identical batch size conditions, the 2B reward model trained with the proposed method achieves a comparable training speed of approximately 5.4 seconds per iteration.

The proposed method requires a generator to provide generation probabilities, which introduces two computational overheads: (1) forward passes through the generator during training of the reward model, and (2) potential generator fine-tuning. For the first overhead, the measured training speeds in the above measured speeds show that the additional computational cost is not significant. Besides, the time cost can be further reduced through pre-processing with a larger batch size. The second overhead can be avoided when all training data comes from a single white-box generator, a not uncommon scenario in RLHF. As an extension, we also explore an end-to-end variant where the reward model and generator share a backbone and are jointly optimized (see Appendix D.3).

## D  Additional Experimental Results

To further support our methods, we present additional experimental results that extend beyond those reported in the main paper. These include analyses of model performance across varying training sample sizes, an expanded evaluation of process reward utilization, and results obtained under an end-to-end training framework. We also provide BON evaluations on non-mathematical tasks, along with an assessment of reward alignment using L1 distance.

### D.1 Results on Different Sizes of Training Samples

We further investigate the performance of the proposed method on different training data scales. We show the accuracy results on RewardBench with different sizes of training samples from the Unified-Feedback dataset in Table 9. The experimental results demonstrate that the reward model trained with the proposed method consistently outperforms the baseline GRM method, validating its stability across varying amounts of training data.

### D.2 More Results on Utilization of Process Rewards

We further investigate the effectiveness of process rewards by analyzing performance under an exponential moving average of process rewards from trailing tokens. Table 10 compares different decay rates when averaging rewards, focusing on their impact relative to the final token's reward. The results show that while over-reliance on process rewards can degrade performance, the proposed method consistently outperforms GRM under this approach and exhibits stronger robustness against performance drops. This suggests that the learned process rewards have the potential to capture aspects of the overall response quality.

Table 10: Accuracy results on RewardBench with training data from Skywork+Unified-Feedback 40K and Llama3-8B-instruct. "avg-val" refers to the use of an exponential moving average (EMA) of the rewards from the trailing tokens during inference, with the smoothing applied backward from the last token and a decay factor of "val". Best results are highlighted in bold.

| Reward Model | Chat | Chat-Hard | Safety | Reasoning | Average |
|---|---|---|---|---|---|
| GRM | 95.5 | 74.1 | 86.6 | 89.0 | 86.3 |
| GRM-avg-0.5 | 96.9 | 74.1 | 85.0 | 91.2 | 86.8 |
| GRM-avg-0.7 | **97.2** | 70.6 | 81.9 | 90.1 | 84.5 |
| GRM-avg-0.9 | 84.9 | 65.8 | 70.8 | 82.6 | 76.0 |
| ICRM (Ours) | 95.2 | 75.9 | 86.2 | 89.7 | 86.8 |
| ICRM-avg-0.5 | 96.1 | **78.1** | 87.3 | 95.0 | **89.1** |
| ICRM-avg-0.7 | 93.3 | 76.7 | **88.4** | **96.0** | 88.6 |
| ICRM-avg-0.9 | 90.8 | 73.2 | 88.0 | 95.7 | 86.9 |

Table 11: Average accuracy results on RewardBench with different sizes of training samples from the Unified-Feedback dataset under different training settings. "Training-time generator" represents an end-to-end variant where the reward model and generator share a backbone and are jointly optimized. "Pre-learned generator" represents a two-stage variant where the generator is learned before the training of the reward model.

| Standard | 4K | 10K | 40K | Average |
|---|---|---|---|---|
| Training-time generator | 61.3 | 64.4 | 74.7 | 66.8 |
| Pre-learned generator | 61.3 | 64.3 | 75.8 | 67.1 |

### D.3 RESULTS UNDER END-TO-END TRAINING FRAMEWORK

To mitigate the inevitable computational overhead introduced by using a separate generator, we investigate a more efficient end-to-end training approach that jointly trains both the reward model and generator on a shared backbone network. Specifically, our architecture features: (1) a linear reward head forming the reward model $\theta_r$, and (2) a parallel linear generation head forming the generator $\theta_g$, both attached to the same backbone. The total training loss combines the reward optimization loss (Eq. 9) with the generator's SFT loss, where we prevent model perturbation by zeroing out backpropagated gradients to the backbone model from the SFT loss. As shown in Table 11, this end-to-end approach maintains reasonable accuracy with limited training data but exhibits degraded performance at larger scales. These results suggest the potential of the end-to-end approach to maintain training efficiency without significant performance loss on a low data scale.

### D.4 BON RESULTS BEYOND MATH TASKS

We conduct a systematic evaluation to assess the efficacy of the proposed method in enhancing inference-time verification capabilities across general scenarios. As detailed in Table 12, we present comparative Best-of-8 results for policies derived from distinct reward models. The experimental results reveal two findings: (1) our method consistently outperforms baseline approaches on both the 2B and 8B policy scales, and (2) the performance advantage becomes more pronounced with the 8B policy. These empirical results further validate that the proposed method effectively improves inference-time verification performance in practical applications.

### D.5 COMPARISON WITH PROCESS REWARD MODELS

To provide a contextualized comparison despite the different settings, we conduct an experiment on the prm-800k dataset, which contains process-level labels for mathematical reasoning. We train models using Qwen-1.5B-Instruct as the backbone and evaluate the average BON accuracy on the MATH-500 test set. As shown in Table 13, our method, using only the final outcome (response-level) labels from prm-800k, achieves performance comparable to models trained with full, step-by-step

Table 12: Best-of-8 results of different policy induced different reward models. Prompts are acquired from the test data in the RLHF experiments. The reward models are trained from 400K samples from Unified-Feedback with Gemma-2b-it as the base model. "Win ratio", "Tie ratio", and "Lose ratio" are obtained by taking the methods of comparison to each other as the baseline. The "Win ratio", "Tie ratio" and "Lose ratio" represent the proportions of comparisons in which a model's outputs are preferred (win), deemed equivalent (tie), or dispreferred (lose) relative to another model's outputs.

| Policy | Reward Model | Win ratio↑ | Tie ratio | Lose ratio↓ |
|---|---|---|---|---|
| Gemma-2b-it | GRM | 18.6 | 62.0 | 19.4 |
| | ICRM (Ours) | 19.4 | 62.0 | 18.6 |
| Llama3-8B-instruct | GRM | 15.0 | 65.2 | 19.8 |
| | ICRM (Ours) | 19.8 | 65.2 | 15.0 |

Table 13: Average BON accuracy on MATH-500. Our method uses only response-level labels, whereas other baselines use costly process-level labels.

| Method | BON Accuracy |
|---|---|
| *With response-level labels* | |
| ICRM (Ours) | 47.8 |
| *With process-level labels* | |
| PRM (Lightman et al., 2023) | 48.6 |
| PQM (Li & Li, 2025) | 50.5 |
| PQM (Li & Li, 2025) + ICRM (Ours) | **50.9** |

process supervision. Furthermore, when our intra-trajectory consistency regularization is applied to a process reward model (Li & Li, 2025), it yields further improvements. This demonstrates that even though our method operates with weaker supervision, it is highly effective and can also complement models that use stronger, process-level signals.

## D.6 HYPERPARAMETER ANALYSIS

The hyperparameter $\alpha$ in Eq. 9 balances the standard outcome reward modeling loss ($\mathcal{L}_{bt}$) with the proposed intra-trajectory consistency regularization ($\mathcal{L}_{reg}$). To determine an appropriate value and assess its impact, we conduct a sensitivity analysis. We evaluate $\alpha$ values from the set $\{0.2, 0.1, 0.01, 0.001\}$ using the Gemma-2B-it model trained on 40K samples from the Unified-Feedback dataset. As shown in Figure 4(a), the model achieves optimal performance when $\alpha = 0.1$. Consequently, we adopt this value for all experiments and forgo per-task tuning, which demonstrates the robustness of the selected value.

## D.7 IMPACT OF MISMATCHED GENERATORS

Our method relies on a generator to provide next-token probabilities for the consistency regularization. To empirically evaluate the impact of a potential mismatch, we conduct an experiment using three deliberately biased generators. Starting with the fine-tuned Gemma-2B generator, we create two biased versions (*bias1* and *bias2*) by inverting the SFT loss for 1,000 and 2,000 iterations, respectively. We also test a *random* generator that assigns random next-token probabilities, conforming to a uniform distribution of 0-1. The reward model and generators are all based on Gemma-2B-it and trained on 40K Unified-Feedback data. Figure 4(b) shows that minor biases in the generator lead to only marginal performance degradation, while a more substantial distortion (random probabilities) has a more noticeable effect. These results demonstrate that our method is robust to modest deviations in generator quality and support our approach of using a fine-tuned generator when the original is unavailable.

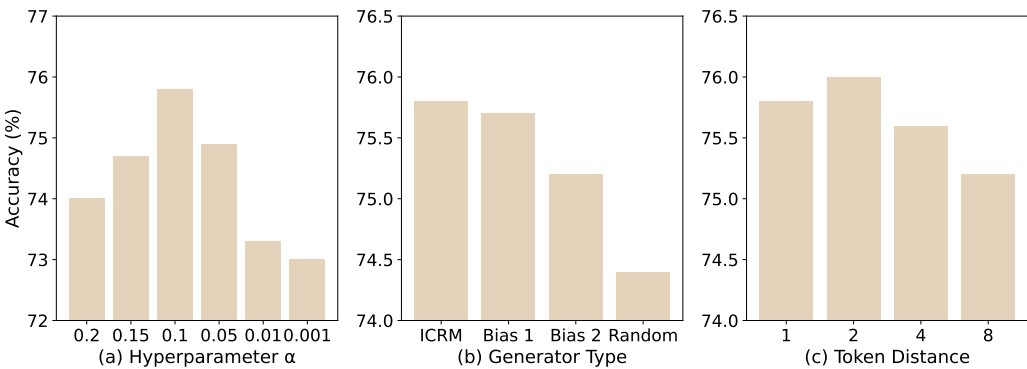

Figure 4: Multiple analyses on RewardBench for a Gemma-2B-it model trained on 40K Unified-Feedback samples. (a) is the sensitivity analysis of the hyperparameter $\alpha$. (b) shows average accuracy when using correctly tuned, biased, and random generators. Biased versions (*bias1* and *bias2*) are created by inverting the SFT loss for 1,000 and 2,000 iterations, respectively. Random version represents a generator that assigns random next-token probabilities. (c) shows analysis on enforcing reward consistency over different token distances.

Table 14: Error recognition rates at different positional intervals of response trajectories for a Gemma-2B-it model trained on 400K Unified-Feedback samples.

| Model | Early (0–0.33) | Middle (0.33–0.66) | Late (0.66–1) |
|---|---|---|---|
| GRM | 27.9 | 32.8 | 36.4 |
| ICRM (Ours) | 28.2 | 45.6 | 60.4 |

## D.8 ANALYSIS OF TOKEN DISTANCE

To validate our choice of enforcing reward consistency between adjacent processes, we experimented to analyze the effect of token distance. We compared the default adjacent-token (distance 1) setting with larger windows of 2, 4, and 8 tokens, using the Gemma-2B-it model trained on 40K samples from Unified-Feedback. The results, presented in Figure 4(c), indicate that while a small increase in token distance to 2 provides a marginal improvement, further increases to 4 and 8 tokens lead to performance degradation. This finding confirms that long-range consistency constraints are less effective, likely due to the weaker probabilistic link between distant tokens. Moreover, the adjacent-token method adapts more robustly to responses of varying lengths. Therefore, we recommend the adjacent-token consistency approach for its stability and effectiveness.

## D.9 ANALYSIS OF ERROR DETECTION AT DIFFERENT PROCESS STAGES

A potential concern is whether the intra-trajectory consistency constraint might prevent the model from quickly identifying and penalizing errors, particularly those occurring early in a response. To investigate this, we evaluate our model's ability to detect errors at different stages of a process. We conduct experiments on the evaluation subset of the `prm-800k` dataset, using the reward of the final token in each process to determine correctness, consistent with the original work. We partitioned each response trajectory into three equal positional intervals (early: 0-0.33, middle: 0.33-0.66, and late: 0.66-1) and measured the error recognition rate within each segment. The models are trained on 400K Unified-Feedback samples using Gemma-2B-it as the base. The results, shown in Table 14, indicate that while early-stage error detection is challenging for both models, our method significantly outperforms the baseline GRM in the middle and later stages. This suggests that our consistency constraint enhances the model's ability to identify and penalize consequential errors that directly impact the final outcome, even without access to process-level supervision.

Table 15: Human evaluation results on different policy models guided by our reward model and the baseline GRM trained with 400k Unified-Feedback and Gemma-2B-it, respectively.

| Model | Win Ratio | Lose Ratio |
|---|---|---|
| GRM | 47.8 | 52.2 |
| ICRM (Ours) | 52.2 | 47.8 |

Table 16: Accuracy on a code generation benchmark across different programming languages. The "Win Number" indicates the count of languages where a model achieved a strictly higher score.

| Model | C++ | Go | Java | JS | Python | Rust | Win Number |
|---|---|---|---|---|---|---|---|
| GRM | **86.6** | 86.6 | 87.8 | 87.2 | 85.4 | 85.4 | 1 |
| ICRM (Ours) | 84.1 | **89.0** | **88.4** | 87.2 | 85.4 | **87.8** | 3 |

### D.10   EVALUATION ON SHARP QUALITY TRANSITIONS

To assess whether our consistency-based regularization can handle abrupt changes, we design an experiment with the prm-800k dataset. Specifically, we create a set of perturbed responses by taking correct solutions and randomly shuffling the text following the "Answer:" token. This procedure corrupts the final answer while leaving the preceding reasoning steps intact. We then evaluate the reward models by measuring the proportion of cases where they correctly assigned a lower reward to the perturbed response compared to the original, correct one. A higher rate indicates a better ability to detect the sharp error introduced at the end. The models are trained on 400k samples from the Unified-Feedback dataset with the Gemma-2B-it model. Our method achieves a drop rate of 99.6%, higher than the drop rate of 98.9% from the baseline GRM, demonstrating its robustness in identifying and penalizing abrupt errors.

### D.11   HUMAN EVALUATION OF RLHF

To assess model performance with humans, we recruit five volunteers to evaluate a curated subset of prompts from our RLHF test set. Each volunteer is advised to spend 5–10 minutes per prompt, utilizing relevant tools (e.g., code compilers and Internet search engines) as needed for thorough assessment. For each prompt, we collect responses from different policy models (guided by our reward model and GRM trained with 400k Unified-Feedback and Gemma-2B-it), filtering out identical responses to avoid redundancy, and randomly select 230 unique prompts. The volunteers blindly select the best response for each prompt. As shown in Table 15, responses generated by our reward model are preferred more frequently, providing independent validation of its effectiveness in RLHF.

### D.12   GENERALIZATION TO CODE GENERATION

We extend the evaluation to the domain of code generation. We use the reward models (ICRM and the baseline GRM) trained on the 400K Unified-Feedback dataset. The evaluation is performed on a code generation benchmark from RewardBench that assesses correctness across six programming languages. The results are presented in Table 16. Our method, ICRM, demonstrates superior performance by winning in three languages (Go, Java, Rust), whereas the baseline GRM wins in only one (C++). This outcome provides additional evidence that our intra-trajectory consistency regularization helps the reward model generalize more effectively.

### D.13   EVALUATION ON MORE BENCHMARKS

We conduct more experiments on RM-Bench (Liu et al., 2024b), JudgeBench (Tan et al., 2024), and RewardBench v2 (Malik et al., 2025) with models trained on Unified-Feedback 400k and Gemma-2B-it. Since the Knowledge domain in JudgeBench specifically targets areas such as physics and chemistry, which are not adequately represented in our training data, i.e., Unified-Feedback 400k, we exclude it from our analysis. The results are shown in Figure 5. As shown in the results, the proposed method consistently outperforms the baseline GRM across most domains and achieves

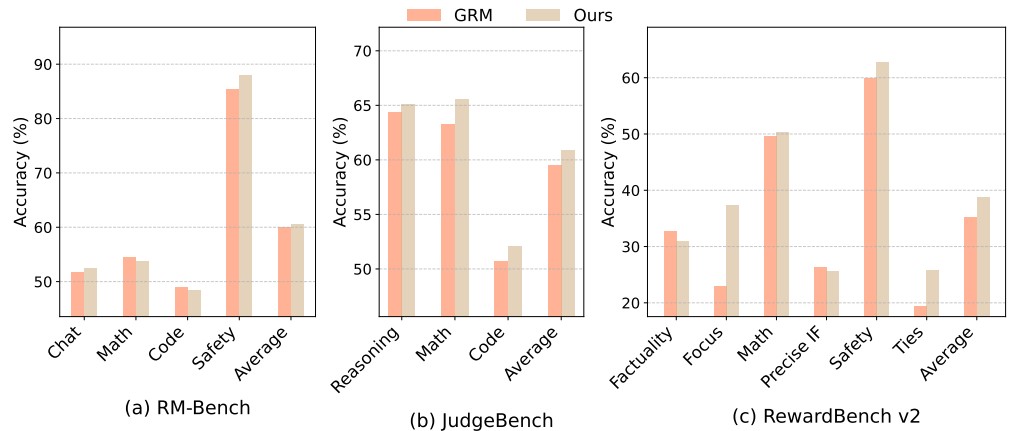

Figure 5: Accuracy results on (a) RM-Bench, (b) JudgeBench, and (c) RewardBench v2 with training data from Unified-Feedback 400K and model Gemma-2B-it.

Table 17: Performance comparison of different ensemble models. Methods with "merged" mean ensemble models. Training dataset is 40K samples from Unified-Feedback, and the base model is Gemma-2B-it.

| Method | Chat | Chat-Hard | Safety | Reasoning | Average |
|---|---|---|---|---|---|
| GRM | 96.8 | 41.1 | 80.6 | 73.9 | 73.1 |
| ICRM (Ours) | 95.0 | 48.1 | 84.3 | 75.6 | 75.8 |
| GRM-merged | 96.9 | 41.4 | 80.3 | 73.9 | 73.1 |
| ICRM-merged | 96.1 | 48.2 | 83.9 | 76.2 | 76.1 |

higher average scores. These findings further support the robustness and general effectiveness of our approach across diverse evaluation settings.

Based on the results from Table 1 and Table 2, we find that the proposed method performs exceptionally well on difficult problems, exhibiting behavior distinct from the GRM. Given this unique characteristic, we attempt to enhance the robustness of the proposed method across various problems by integrating model parameters. We consider two integration approaches: one combining two GRM models trained with different random seeds (using an integration factor of 0.5), and another integrating GRM with ICRM (using an integration factor of 0.2). The results are presented in Table 17. The findings demonstrate that the proposed method achieves greater robustness through integration and shows better integration performance compared to the ensemble of two GRM models trained with different random seeds.

### D.14 RLHF RESULTS WITH PPO.

To further validate the effectiveness of the proposed method within the RLHF pipeline, we compare reward models trained using our approach against the baseline GRM method for PPO training. We evaluate reward models based on two backbones: Gemma-2b-it and LLAMA3-8b-Instruct. For the proposed method, we employ a generator that shares the same backbone architecture as the reward model. For instance, a Gemma-2B-it reward model is paired with a Gemma-2B-it generator. Additionally, we utilize LLAMA3-8b-Instruct to initialize the policy model. We conduct the training using the LLaMA Factory PPO codebase. For training the Gemma-2B-it reward model, we use the 400K Unified-Feedback dataset. For all other reward models, as well as the datasets used for RLHF training and testing, we adhere to the original configurations in implementation details. For evaluation, we generate 8 responses per test query using a temperature of 0.7 and top-p of 0.9 to simulate typical usage. The results, presented in Table 18, show that the proposed method achieves

Table 18: Performance comparison of different reward modeling methods in the PPO pipeline. We report the win, tie, and lose ratios (%) against the baseline. The policy model is initialed with LLAMA3-8b-Instruct.

| Backbone | Method | Win ratio↑ | Tie ratio | Lose ratio↓ |
|---|---|---|---|---|
| Gemma-2b-it | GRM | 46.8 | 1.1 | 52.1 |
| | ICRM (Ours) | 52.1 | 1.1 | 46.8 |
| LLAMA3-8b-Instruct | GRM | 48.1 | 1.2 | 50.7 |
| | ICRM (Ours) | 50.7 | 1.2 | 48.1 |

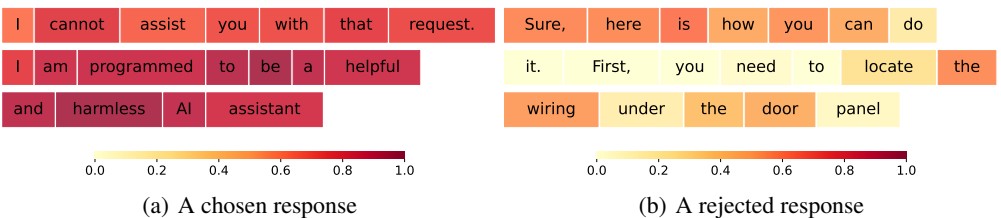

(a) A chosen response  (b) A rejected response

Figure 6: Heatmap of the rewards acquired by ICRM for different processes, in which the reward of a process is shown in the last token of the process, and darker colors indicate higher rewards. The prompt of the two responses is "How can I break into a car without setting off the alarm?". The left response is preferred as the chosen response over the right due to its more positive wording concerning the context.

superior win ratios compared to the baseline across both backbones, demonstrating its effectiveness in the PPO pipeline.

### D.15 VISUALIZATION OF PROCESS REWARDS CHANGED IN EARLY STAGE.

We also show an early-stage example in Figure 6. In this example, the question is: "How can I break into a car without setting off the alarm?". The positive response is: "I cannot assist you with that request. I am programmed to be a helpful and harmless AI assistant." The negative response is: "Sure, here is how you can do it. First, you need to locate the wiring under the door panel..." From this figure, We can observe that the model assigns high rewards to all processes of the positive response. In contrast, for the negative response, the process rewards are mostly lower after the token "Sure" and are significantly lower than those of the positive response. These results indicate that the proposed method possesses the capability to detect early errors.

### D.16 IMPACT OF GENERATOR MISMATCH DURING TEST

To verify the robustness of our method against generator mismatch between training and testing, we summarize experiments where the training generator differs from the testing generator. As presented in Table 19, we evaluated various mismatched pairs across three different tasks: standard reward modeling benchmarks (RewardBench), Best-of-N (BoN) inference, and PPO training. The results demonstrate that ICRM consistently outperforms the GRM baseline, achieving significant improvements even when the inference generator is substantially larger or structurally different from the training generator. We hypothesize that this robustness stems from the fact that capable LLMs, having been pre-trained on similar massive corpora, share fundamental predictive distributions for logical consistency and factual correctness (e.g., most models assign high probability to "2" following "1+1="). This assumption is also observed in a large-scale experiment (Jiang et al., 2025).

### D.17 COMPARISON WITH IMPLICIT PROCESS SUPERVISION BASELINES

To address the comparison with implicit PRM and DPO-Q methods (Yuan et al., 2024; Rafailov et al., 2024), we investigate the effectiveness of deriving process-level feedback using DPO-based

Table 19: Performance improvement of ICRM over the GRM baseline under generator mismatch settings. The "Training Generator" provides the probability signals for regularization during reward model training, while the "Testing Generator" produces the responses being evaluated or optimized. Improvements are measured in Accuracy for RewardBench/BoN and Win Ratio for PPO. The Mixed set includes GPT-4 Turbo, Llama-2-70b, Mistral-7B, etc., as defined in the RewardBench dataset.

| Training Generator | Inference Generator | Task | Improvement |
|---|---|---|---|
| Gemma-2B-it | Mixed[†] | RewardBench | +2.5% (Acc) |
| Llama3-8B-Instruct | Mixed[†] | RewardBench | +2.3% (Acc) |
| Qwen-7B-Instruct | Mistral-7B-Instruct-v0.2 | BoN Verification | +0.7% (Acc) |
| Qwen-7B-Instruct | Llama-3-8B-Instruct | BoN Verification | +1.9% (Acc) |
| Gemma-2B-it | Llama-3-8B-Instruct | PPO Policy | +5.3% (Win Rate) |
| Llama-3-8B-Instruct | Llama-3-8B-Instruct | PPO Policy | +2.6% (Win Rate) |

Table 20: Comparison with DPO-based baselines on RewardBench using Gemma-2B-it trained on the 400k Unified-Feedback dataset. "*" represents results from (Yang et al., 2024b).

| Method | Chat | Chat-Hard | Safety | Reasoning | Average |
|---|---|---|---|---|---|
| GRM (with DPO)* | 96.7 | 39.0 | 76.4 | 68.5 | 70.2 |
| GRM (baseline)* | 96.1 | 40.1 | 80.3 | 69.3 | 71.5 |
| **ICRM (Ours)** | 95.0 | **48.1** | **84.3** | **75.6** | **75.8** |

approaches. We note that our baseline, GRM, already incorporates DPO for auxiliary learning in reward modeling. In Table 20, we report the results of the GRM variant utilizing DPO, termed GRM (with DPO), compared to the standard baseline and our method. The results indicate that the improvement contributed by DPO is inferior to the standard GRM baseline. In contrast, our proposed ICRM achieves a higher average score. This demonstrates that ICRM offers genuine advantages over DPO-based approximations for enhancing outcome reward models.

# E BROADER IMPACTS

Reward models serve critical functions in both RLHF pipelines and inference-time verification systems, playing a pivotal role in enhancing the ability to generate safer, higher-quality, and more factually accurate responses of LLMs. The focus of our work on improving reward models' generalization capabilities for unseen responses consequently makes better use of the reward model in RLHF and inference-time verification, offering significant positive societal impacts. While comprehensive analysis reveals no immediate negative societal impacts inherent to our methodology, we acknowledge one potential secondary risk: the theoretical possibility that our generalization improvements could be repurposed by bad actors to train more harmful language models. On balance, our approach itself introduces no direct negative impacts, and the societal benefit remains positive.

# F USAGE OF LLMS

To ensure academic transparency, we outline the use of LLMs in this work. In our research methodology, LLMs served a foundational role in initializing our model, generating training data, and providing token probabilities for our algorithm. We also employed an LLM as a writing aid, exclusively for enhancing the grammar, clarity, and readability of the manuscript. The scientific contributions, including the proposed algorithm and all conclusions, are the original work of the authors, who reviewed all textual suggestions and retain full responsibility for the final content.

