# OpenReview forum: "Intra-Trajectory Consistency for Reward Modeling"
_ICLR.cc/2026/Conference — Submitted to ICLR 2026_

### Official Review · Reviewer_qRsQ · 2025-10-23

**Soundness:** 2
**Presentation:** 2
**Contribution:** 2
**Rating:** 2
**Confidence:** 3

**Summary:**

The paper proposes an Intra-Trajectory Consistency Regularization (ICRM) method for reward modeling, aiming to improve credit assignment along reasoning trajectories without requiring explicit step-level labels. The key idea is to enforce consistency between neighboring reasoning steps by introducing a probabilistic regularization term, which supposedly yields more coherent and fine-grained process rewards. The authors claim that ICRM outperforms generalized reward models (GRMs) and enhances downstream reasoning tasks such as best-of-N reranking and RLHF-style fine-tuning.

**Strengths:**

1. The work addresses a meaningful problem in reward modeling, how to approximate process-level supervision from outcome-level signals, which is timely and important for reasoning-heavy LLM tasks.
2. The idea of linking neighboring steps via a Bayesian decomposition is conceptually interesting and could potentially bridge ORM and PRM methods.
3. The authors evaluate across multiple tasks and models, using RewardBench and RLHF-style setups.
4. The paper provides ablations and partial visualizations (heatmaps, section D) that help interpret the proposed regularization.

**Weaknesses:**

1. Main claim not empirically supported: The paper claims that ICRM improves credit assignment along trajectories, but the presented heatmaps primarily reflect accumulative error effects (probabilities naturally decay toward later tokens in autoregressive models). Without disentangling this from the regularizer’s effect, it is unclear whether credit assignment actually improved.

2. Visualization is confounded by probability decay: Figure 3’s token-level heatmap shows reward decline mainly in later segments, which likely results from accumulated log-probability decay rather than effective intra-step credit redistribution. No analysis isolates the effect of the regularizer from this autoregressive bias.

3. Selective reporting and overstatements: Claims such as “ICRM surpasses all GRMs” are overstated. Tables show categories where GRM still performs better (e.g., Chat domain). The improvements are average, not universal.

4. Lack of statistical rigor: Standard deviations and multiple random seeds are missing for most key metrics. RLHF evaluations rely solely on a single automatic judge (QRM-Llama3.1) rather than human ratings, making results less reliable.

5. Credit signal concentrated at sequence end: Section D.10 shows that ICRM primarily improves error detection in later parts of the trajectory, with little gain in early steps. This suggests that the method strengthens terminal penalties rather than distributing credit more effectively throughout the process.

6. No control for probability influence: Since the regularization explicitly uses next-token probability as its weighting factor, the correlation between probability and reward consistency should be empirically verified. The authors provide no statistical analysis (e.g., correlation plots or regression residuals) to support the assumed relationship.

7. Weak comparison baseline: The paper only compares against GRMs, but recent implicit PRM and DPO-Q works (e.g., Yuan et al., 2024; Rafailov et al., 2024) already achieve process-level supervision from outcome labels. Without including those baselines, it is unclear whether ICRM offers any genuine advantage.

8. Interpretation inconsistencies: The authors attribute posterior reward decay to “better credit distribution,” yet the data could equally reflect probability mass shrinkage or end-of-sequence effects. This ambiguity is never addressed.

**Questions:**

1. Can you provide an analysis separating the influence of next-token probability from the effect of consistency regularization? For instance, show residual rewards after regressing out token probabilities.
2. How does ICRM perform if the regularizer is disabled but probabilities are still normalized? Does the improvement persist?
3. Why are there no standard deviations or multiple random seeds reported for the main tables?
4. Have you compared ICRM directly against implicit PRM or DPO-based Q-value approximations, which also derive process-level feedback from outcome labels?
5. Can you visualize cases with early or mid-sequence errors rather than only late errors to test whether ICRM truly improves step-level credit assignment?
6. Does the observed performance improvement hold when the generator and reward model come from different distributions or architectures?

---

> ### Author Response · Authors · 2025-11-22
>
> We thank Reviewer qRsQ for recognizing the significance of our method, the comprehensiveness of our experimental validation, and the interpretability of our results analysis. And we address the concerns below.
>
> ---
>
> Before addressing specific questions, we would like to clarify the primary objective of our method and certain technical concepts regarding our framework.
>
> (1) Clarification on Research Objective. The goal of our method is to introduce regularization signals via process consistency to enhance the accuracy of the outcome reward. Therefore, even if our method does not achieve perfect process reward learning, the error recognition rates across different position stages (Table R1) show that the proposed method provides significantly more accurate process rewards than standard ORM, which can validate the potential of the proposed method to improve ORM. Extensive experimental results in the paper further confirm that our method indeed improves the accuracy of outcome rewards.
>
> *Table R1: Error recognition rates at different positional intervals of response trajectories.*
>
> |          | 0-0.33 (early stage) | 0.33-0.66 (middle stage) | 0.66-1 (late stage) |
> | -------- | ------ | --------- | ------ |
> | GRM | 27.9   | 32.8      | 36.4   |
> | ours     | 28.2   | 45.6      | 60.4   |
>
>
> (2) Clarification on Technical Concepts. We need to emphasize that we have not employed the concept of ''credit assignment'' in any position of the paper. We guess the reviewer may refer to the concept of ''process reward'' as ''credit assignment''. Therefore, we also need to clarify the distinction between process rewards and generation probabilities. In our method, both outcome and process rewards are derived via a binary classifier, which is fundamentally different from the multi-label generation probabilities. As illustrated in Figure 1 in the paper, the source of the reward signals is distinct from the generation probabilities. Consequently, during the training of the reward model, we do not train the generation probability component (the generator), nor do we use generation probabilities to generate the process rewards. Our reported improvements strictly reflect the enhanced discriminative capability of the reward model, not an improvement in the generator's prediction accuracy.

---

> > ### Author Response · Authors · 2025-11-22
> >
> > ---
> >
> > # Response to W1, W2, and W8
> >
> > **W1: Main claim not empirically supported: The paper claims that ICRM improves credit assignment along trajectories, but the presented heatmaps primarily reflect accumulative error effects (probabilities naturally decay toward later tokens in autoregressive models). Without disentangling this from the regularizer’s effect, it is unclear whether credit assignment actually improved. W2: Visualization is confounded by probability decay: Figure 3’s token-level heatmap shows reward decline mainly in later segments, which likely results from accumulated log-probability decay rather than effective intra-step credit redistribution. No analysis isolates the effect of the regularizer from this autoregressive bias. W8: Interpretation inconsistencies: The authors attribute posterior reward decay to “better credit distribution,” yet the data could equally reflect probability mass shrinkage or end-of-sequence effects. This ambiguity is never addressed.**
> >
> > Thank you for your review. Figure 3 indeed demonstrates effective process reward allocation rather than merely reflecting the cumulative effect of generation probabilities. Specifically, in the rejected response (b), we observe a distinct drop in reward scores at semantically negative terms (e.g., "ignore" and "cruel words"), while the rewards for preceding tokens (e.g., "is to") remain relatively high. In contrast, the chosen response (a) maintains high reward levels throughout the trajectory (including "respect" and "kind words"), significantly outscoring the negative terms in (b). If these results are solely driven by cumulative probability decay, both responses would exhibit similar decay patterns. This divergence indicates that our model has learned to associate rewards (or penalties) with specific, meaningful tokens, rather than blindly decaying process rewards over length. Furthermore, we clarify that the improvement claimed by our method refers to the enhancement of process rewards relative to the ORM baseline, not an improvement in generation probabilities relative to the reference generation model. This statement is further supported by original paper (Appendix D.10, Page 19). For convenience, we also present the results in the following table (Table R1), which shows that our method achieves a significantly higher average error recognition rate compared to typical outcome reward modeling method GRM.
> >
> > ---
> >
> > # Response to W3
> >
> > **Selective reporting and overstatements: Claims such as “ICRM surpasses all GRMs” are overstated. Tables show categories where GRM still performs better (e.g., Chat domain). The improvements are average, not universal.**
> >
> > Thank you for your feedback. We have revised the wording in the paper to more accurately state that: ICRM demonstrates superior average performance over the GRM baseline on the benchmark, with notable improvements particularly in more challenging categories such as "Chat-Hard" and "Reasoning".

---

> ### Author Response · Authors · 2025-11-22
>
> ---
>
> # Response to W4.1 and Q3
>
> **W4.1: Lack of statistical rigor: Standard deviations and multiple random seeds are missing for most key metrics. Q3: Why are there no standard deviations or multiple random seeds reported for the main tables?**
>
> Thank you for your review. In the original manuscript (Table 1, Page 6), we have conducted an analysis study of standard deviations. For convenience, we also present the results in the following table (Table R2). We also performed statistical significance testing on these results, yielding a p-value of 0.002, which confirms the significance of our improvements. Given the high computational costs associated with LLM experiments and the fact that Table R2 already demonstrated the statistical significance of our method's gains, we did not complete multiple seed runs for every table at the time of submission. However, we will endeavor to supplement these data in the final version to further enhance statistical rigor.
>
> *Table R2: Average accuracy and standard deviation on RewardBench.*
>
> | | mean | std |
> |-|-|-|
> | GRM | 73.1 | ±0.2 |
> | ours | 75.8 | ±0.4 |
> | p value | 0.002 |
>
>
> ---
>
> # Response to W4.2
>
> **RLHF evaluations rely solely on a single automatic judge (QRM-Llama3.1) rather than human ratings, making results less reliable.**
>
> Thank you for your review. In the original manuscript (Appendix D.12, Page 20), we have conducted an analysis study of human evaluation. For convenience, we also present the results in the following table (Table R3). To assess model performance with humans, we recruit five volunteers to evaluate a curated subset of prompts from our RLHF test set. Each volunteer is advised to spend 5–10 minutes per prompt, utilizing relevant tools (e.g., code compilers and Internet search engines) as needed for thorough assessment. For each prompt, we collect responses from different policy models (guided by our reward model and the baseline GRM trained with 400k Unified-Feedback and Gemma-2B-it), filtering out identical responses to avoid redundancy. The volunteers blindly select the best response for each prompt. As shown in the table below, responses generated by our reward model are preferred more frequently, providing independent validation of its effectiveness in RLHF.
>
> *Table R3: Human evaluation results on different policy models guided by different reward models.*
>
> |          | Win ratio | Lose ratio |
> | -------- | --- | ---- |
> | GRM |   47.8  |   52.2   |
> | ours     |  52.2   |    47.8  |
>
> ---
>
> # Response to W5
>
>
> **Credit signal concentrated at sequence end: Section D.10 shows that ICRM primarily improves error detection in later parts of the trajectory, with little gain in early steps. This suggests that the method strengthens terminal penalties rather than distributing credit more effectively throughout the process.**
>
> We appreciate the reviewer's examination. However, the conclusion that "signals are concentrated only at the end" is a misinterpretation of the data in Table 14 from the original paper. The table (also listed in Table R1) explicitly demonstrates a significant and substantial improvement in error detection rates for ICRM in the "Middle" stage (0.33–0.66), increasing from 32.8\% (GRM) to 45.6\% (ICRM). Furthermore, as clarified at the beginning of this rebuttal, even if our method has not achieved perfect error recognition capability, the improvement relative to the ORM baseline (GRM) effectively demonstrates the potential of the proposed method to enhance ORM methods.

---

> ### Author Response · Authors · 2025-11-22
>
> ---
>
> # Response to W6, Q1, and Q2
>
> **W6: No control for probability influence: Since the regularization explicitly uses next-token probability as its weighting factor, the correlation between probability and reward consistency should be empirically verified. The authors provide no statistical analysis (e.g., correlation plots or regression residuals) to support the assumed relationship. Q1: Can you provide an analysis separating the influence of next-token probability from the effect of consistency regularization? Q2: For instance, show residual rewards after regressing out token probabilities. How does ICRM perform if the regularizer is disabled but probabilities are still normalized? Does the improvement persist?**
>
> Thank you for your question. In the original manuscript (Table 6, Page 8), we have conducted an analysis study of human evaluation. For convenience, we also present the results in the following table (Table R4). In the "w/o adjacent reg" experiment, we remove the mutual weighting derived from adjacent process rewards (i.e., the consistency component) while retaining the probability weighting. This results in a performance drop from 75.8\% to 73.5\%. In the "w/o generation reg" experiment, we remove the weighting based on generation probabilities but retain the logic of adjacent consistency. Here, performance declines from 75.8\% to 75.2\%. Collectively, these results demonstrate that both mechanisms are indispensable components of our proposed method.
>
> *Table R4: Ablation study of the proposed method.*
>
> | Method             | Chat | Chat-Hard | Safety | Reasoning | Average |
> |--------------------|------|-----------|--------|-----------|---------|
> |w/o adjacent reg    |96.1  |43.2       |80.8    |74.0       | 73.5    |
> | w/o generation reg | 95.2 | 46.9      | 83.5   | 75.2      | 75.2    |
> | Overall            | 95.0 | 48.2      | 84.2   | 75.8      | 75.8    |
>
>
> ---
>
> # Response to W7 and Q4
>
> **W7: Weak comparison baseline: The paper only compares against GRMs, but recent implicit PRM and DPO-Q works (e.g., Yuan et al., 2024; Rafailov et al., 2024) already achieve process-level supervision from outcome labels. Without including those baselines, it is unclear whether ICRM offers any genuine advantage. Q4: Have you compared ICRM directly against implicit PRM or DPO-based Q-value approximations, which also derive process-level feedback from outcome labels?**
>
> Thank you for your suggestion. We acknowledge that DPO possesses the capability to model process. However, the extent to which it enhances ORM methods remains open to discussion. Since our baseline method, GRM [1], also incorporates DPO for auxiliary learning in reward modeling, we report the results from the original GRM paper using the Gemma-2B-it model and the 400k Unified-Feedback dataset below (Table R5) for analysis. These results indicate that the improvement contributed by DPO is inferior to the performance of the GRM version we employed as our baseline. In contrast, our method achieves further improvements over GRM, thereby demonstrating the effectiveness of the proposed approach
>
> *Table R5: Accuracy results on RewardBench.*
>
> |  Method | Chat   | Chat-hard | Safety | Reasoning | Avg   |
> |-----------------|--------|-----------|--------|-----------|-------|
> | GRM (with dpo)  | 96.7   | 39.0      |  76.4  |  68.5     | 70.2  |
> |  GRM (our comparison) | 96.1   | 40.1      | 80.3   |  69.3     | 71.5  |
> | GRM (with ours) |  95.0  | 48.1      | 84.3   |  75.6     | 75.8  |
>
> [1] Regularizing hidden states enables learning generalizable reward model for LLMs. Neurips 2024.
>
>
> ---
>
> # Response to Q5
>
> **Can you visualize cases with early or mid-sequence errors rather than only late errors to test whether ICRM truly improves step-level credit assignment?**
>
> Thank you for the question. In the revised paper (Figure 6, Page 22), we have additionally supplemented an early-stage example. In this example, the question is: "Question: How can I break into a car without setting off the alarm?". The positive response is: "I cannot assist you with that request. I am programmed to be a helpful and harmless AI assistant." The negative response is: "Sure, here is how you can do it. First, you need to locate the wiring under the door panel..." We observe that the model assigns high rewards to all processes of the positive response. In contrast, for the negative response, the process rewards are mostly lower after the token "Sure" and are significantly lower than those of the positive response. These results indicate that the proposed method possesses the capability to detect early errors to some extend.

---

> ### Author Response · Authors · 2025-11-22
>
> ---
>
> # Response to Q6:
>
> **Does the observed performance improvement hold when the generator and reward model come from different distributions or architectures?**
>
> Thank you for the question. (1) In the original manuscript (Appendix D.8, Page 18), we have conducted an analysis study of the generator mismatch. For convenience, we also present the results in the following table (Table R6). To evaluate the impact of generator bias, we conducted experiments using three mismatched generators: bias1 and bias2 (created by inverting the fine-tuning loss after 1000 and 2000 iterations, respectively) and random (which assigns random next-token probabilities). Both the generator and the reward model are trained using Gemma-2B-it and 40K Unified-Feedback data. The results show that small biases lead to only minor performance degradation, while more substantial distortions have a larger effect. These results support our hypothesis and demonstrate that our method is robust to modest deviations in the generator.
>
> *Table R6: Average accuracy when using correctly tuned, biased, and random generator*
>
> | ours | bias 1 | bias 2 | random |
> | ---- | ------ | ------ | ------ |
> | 75.8 | 75.7   | 75.2   | 74.4   |
>
>
> (2) Besides, in the original manuscript (Table 5, Page 8), we have also reported the results when generator and reward model is different. We also present the results in the following table (Table R7). Specifically, the reward model employs the Qwen2.5-1.5B-instruct architecture, while the generator utilizes the Qwen2.5-7B-instruct architecture. After training, we evaluate the inference-time verification capability of the reward model across different policies. The results show that the proposed method achieves a higher average accuracy compared to GRM.
>
> *Table R7: BON results of responses from different polices*
>
> | Policy                   | Reward Model | Pass@2 | Pass@4 | Pass@8 | Pass@16 | Average |
> |--------------------------|--------------|--------|--------|--------|---------|---------|
> | Mistral-7B-Instruct-v0.2 | GRM          | 11.8   | 11.8   | 12.6   | 14.2    | 12.6    |
> |                          | ICRM (Ours)  | 11.8   | 12.8   | 14.6   | 14.0    | 13.3    |
> | Llama-3-8B-Instruct      | GRM          | 45.6   | 46.8   | 49.2   | 45.0    | 46.7    |
> |                          | ICRM (Ours)  | 45.8   | 47.2   | 51.2   | 50.0    | 48.6    |

---

> ### Comment · Reviewer_qRsQ · 2025-11-25
>
> Thank you for the detailed rebuttal. I appreciate the clarifications and the additional tables you provided. However, after carefully reviewing the response, I find that most of my core concerns remain insufficiently addressed. I therefore keep my overall score.
>
> **1. Heatmap Interpretation and Probability Confounding**
>
> The rebuttal argues that Figure 3 reflects meaningful process reward allocation rather than accumulated probability decay, pointing to specific tokens such as “ignore” or “cruel words”. However, this remains an anecdotal example, not a systematic analysis. The concern I raised was not whether one or two hand-picked sequences can show semantic drops, but whether the global pattern of reward decay can be distinguished from autoregressive probability decay near the end of sequences, or	semantic deterioration coinciding with probability deterioration.
>
> The rebuttal does not provide residual analysis after regressing out token probabilities, correlation or independence tests between token log-prob and reward, controlled examples where probability patterns are matched but semantics diverge, or early/mid-position systematic tests beyond a single curated example.
>
> Therefore, the core question remains unanswered:
> Has the method truly improved process rewards, or are the observed heatmap patterns still dominated by autoregressive probability decay?
>
> **2. Position-Wise Error Detection**
>
> The rebuttal highlights improvements in the middle segment of Table R1. While this does show relative gains, the absolute early-stage detection rate remains extremely low (28.2 \%). This supports my original concern: the method appears to concentrate reward signals toward later positions. The new example in the revision is helpful but still anecdotal rather than statistical. Without broader positional analysis, I cannot conclude that early and mid-sequence credit signals have meaningfully improved.
>
> **3. Probability Influence and Regularization Effects**
>
> My earlier request was for explicit disentanglement between probability-based weighting and adjacent-step consistency weighting.
>
> The ablations in Table R4 show that removing either component reduces performance. However, this does not demonstrate how strongly reward behavior is driven by probability vs. consistency, whether reward patterns remain after controlling for probability, or whether the learned reward shape is meaningful beyond probability-shaped biases. The response repeats the ablation numbers but not the type of statistical independence analysis needed to substantiate the claims.
>
> **4. Comparison to Implicit PRM and DPO-Q Methods**
>
> The rebuttal compares against a specific DPO configuration used within GRM, but not the implicit PRM or DPO-as-Q-function variants that more directly match the problem setting (process-level inference derived from outcome labels). Since these works already show strong step-level supervision emerging from outcome-only training, they remain important baselines. Without direct comparison, it is still unclear whether ICRM provides a distinctive advantage in this space.
>
> **5. Statistical Rigor**
>
> While Table R2 offers averages and deviations for one benchmark, the majority of core tables still lack: multi-seed evaluations, confidence intervals, or variance reporting. Given the known volatility of small reward models and the sensitivity of RLHF evaluation pipelines, the statistical foundation remains too weak to shift my assessment.
>
> **6. Human Evaluation**
>
> The additional human evaluation table is appreciated, but the effect size is small (52.2 \% vs 47.8 \%), and the number of prompts is unclear. Without sample size or significance tests, it is difficult to interpret the reliability of these results.
>
> The rebuttal acknowledges several issues and offers clarifications, but the key scientific concerns are not resolved (particularly those regarding probability confounding, lack of true process-level validation, missing baselines, and insufficient statistical analysis). The methodological contributions remain interesting, but the evidence is not yet strong enough to support the claims made in the paper.

---

> ### Author Response · Authors · 2025-11-27
>
> # Response to 1 and 3
>
> **Heatmap Interpretation, Probability Confounding,  Probability Influence, and Regularization Effects**
>
> Thanks. To begin with, we would like to clarify that the core objective of our method is to introduce process reward regularization to enable the model to focus on the processes that influence the outcome reward, thereby improving the accuracy of the outcome reward, rather than achieving the best possible process reward results. Optimal process reward learning requires the ability to identify any process error within a response, even if the complete response is correct or overall superior, which is not entirely identical to our objective. Following the objective, our main experiments and ablation studies are designed to verify improvements in outcome reward accuracy. In this metric, our method consistently outperforms the baseline and aligns with our research objective. While the comparison of "process reward" to the autoregressive model does not directly correspond to this objective.
>
> Regarding the concern that the observed process rewards are merely artifacts of "autoregressive probability decay", we point out that the assumption of natural probability decay is not universally applicable. In our analysis of the Gemma-2b-it with the PRM-800k test set, a dataset used in our paper to detect step-wise errors, we observe that the model's generation probability generally increases as the sequence lengthens, regardless of whether the sample is correct or incorrect. This trend contradicts the hypothesis that probabilities naturally decay toward later tokens. Therefore, the distinct reward drops observed in our heatmaps and improvement in error recognition rates cannot be attributed to the probability decay, but rather reflect meaningful rewards.
>
> *Table R1: Generation probability on PRM-800k for different positional intervals of response trajectories.*
>
> |                              | 0-0.33 | 0.33-0.66 | 0.66-1.0 |
> |------------------------------|--------|-----------|----------|
> | probability in incorrect response | 47.9   | 58.9      | 68.5     |
> | probability in correct response | 55.9   | 68.8      | 73.5     |
>
> # Response to 2
>
> **Position-Wise Error Detection**
>
> Thanks. We would like to emphasize that the primary objective of our method is to utilize process consistency as a regularization term to enhance the accuracy of the outcome reward, rather than to train a perfect process reward model capable of identifying every step-level error. In this context, the lower error recognition rate in the early stages does not constitute a fundamental drawback. On the contrary, the significant performance gains observed in the middle and late stages should be regarded as a distinct advantage. Since a process reward essentially measures the validity of a trajectory from the initial token up to the current step, the correctness of processes in the middle and later stages is intrinsically more correlated with the quality of the final complete response. By successfully guiding the model to attend to these decisive middle-to-late stage processes, our method ensures that the model focuses more on the trajectory components that significantly impact the outcome reward. This behavior is aligned with our design goal of improving outcome-level generalization and accuracy.
>
> # Response to 4
>
> **Probability Influence and Regularization Effects**
>
> Thanks. Regarding the comparison with Implicit PRM and DPO-Q methods, we need to point out that these approaches rely on the DPO loss as their primary optimization objective. Therefore, comparing our method against DPO effectively serves as a direct comparison against these methods. In our previous response, we demonstrated that our method outperforms the "GRM + DPO" baseline, which constitutes a fair and direct comparison of regularization effectiveness. Furthermore, regarding DPO's specific capability to assess response quality (outcome reward accuracy), we recommend that the reviewer refer to comprehensive evaluation frameworks such as RewardBench [1]. Existing benchmarks indicate that DPO usually lags behind classifier-based methods (such as GRM). Since our primary objective is to enhance outcome reward accuracy, benchmarking against and improving upon the stronger, classifier-based paradigm is a more rigorous and reasonable experimental setting.
>
> [1] Lambert, Nathan, et al. Rewardbench: Evaluating reward models for language modeling. Findings of NAACL, 2025.

---

> > ### Author Response · Authors · 2025-11-27
> >
> > # Response to 5
> >
> > **Statistical Rigor**
> >
> > Thanks. We note that due to the prohibitive computational costs associated with LLM training, it is standard practice in many published works to report single-run results, particularly for larger models. To address the concern for rigor within a feasible scope, we implemented repeated experiments on the smaller model (Gemma-2b-it). As reported in Table R2, our method demonstrates a statistically significant improvement with a p-value of 0.002 (< 0.05), confirming that the performance gains are significant. Therefore, given these significant results and the high cost of training LLMs, we do not conduct further experiments on a larger scale.
> >
> > # Response to 6
> >
> > **Human Evaluation**
> >
> > Thanks. Regarding the human evaluation in RLHF, we clarify that a 52.2\% vs 47.8\% win rate is a substantial margin in RLHF. In recent work, which is published in Neurips 2025 [2], shows that an improvement of 1\%-2\% on the RLHF experiment is significant. Besides, our study utilized a sample size of 230 unique prompts, which were randomly selected from the test set in the RLHF experiment after filtering out prompts with identical responses.
> >
> > [2] Ye W, Zheng G, Zhang A. Rectifying Shortcut Behaviors in Preference-based Reward Learning. Neurips, 2025.

---

### Official Review · Reviewer_4pah · 2025-10-26

**Soundness:** 1
**Presentation:** 1
**Contribution:** 2
**Rating:** 2
**Confidence:** 4

**Summary:**

This paper introduces an intra-trajectory consistency regularization method to refine reward models by propagating coarse-grained, response-level supervision into fine-grained learning signals using a Bayesian-inspired principle. The approach improves performance on RewardBench and enhances DPO-aligned policies.

**Strengths:**

1. The theoretical analysis effectively supports the arguments.
2. Introducing token-wise information during the reward model training phase demonstrates innovation.

**Weaknesses:**

1. The additional computational overhead introduced during training is non-negligible. Training a 2B reward model along with a 2B generator should be compared with an ablation study involving training a standalone 3B–4B reward model for a more appropriate evaluation.
2. Generator mismatch is a common issue. On one hand, during RLHF, the reward model size may be significantly smaller than the actor model. On the other hand, the distribution of the actor model can shift considerably as training progresses. Although the authors conducted related experiments in the Appendix, I believe these are insufficient and require more ablation studies involving different model sizes, model series, and training steps.
3. The baselines are overly simplistic. Many methods exist for enhancing reward models, and GRM is already over a year old. I suggest including more baselines for comparison.

**Questions:**

1. Why do larger training sample sizes lead to worse performance in Table 1 and Table 2? I also checked the original GRM paper, and the data presented there differs from what is shown in your paper. Is there an explanation for this discrepancy?
2. Why use DPO? The main purpose of using a BT model is to address reward modeling issues in PPO. For algorithms like DPO, which only require preference information, directly training on the original preference dataset would be more straightforward. Introducing an intermediate reward model seems redundant and may add noise, which I find confusing. I recommend adding experiments with PPO. Moreover, your baseline GRM has been evaluated on PPO and BoN, not DPO.

---

> ### Author Response · Authors · 2025-11-22
>
> We would like to thank Reviewer 4pah for recognizing our method's theoretical contribution and innovative approach. And we address the concerns below.
>
> ---
>
> # Response to W1
>
> **The additional computational overhead introduced during training is non-negligible. Training a 2B reward model along with a 2B generator should be compared with an ablation study involving training a standalone 3B–4B reward model for a more appropriate evaluation.**
>
> Thank you for your valuable review. To begin with, we would like to clarify a misunderstanding regarding the training process. We do not train the generator and the reward model simultaneously. As stated in the Introduction of the paper, the generator in our framework is completely frozen. Therefore, comparing "training a 2B reward model + a frozen 2B generator" to "training a standalone 3B-4B reward model" is not an equivalent comparison. Training a 3B-4B model requires computing gradients (backward pass) for all parameters, which is significantly more expensive than the inference-only forward pass required for our frozen generator.
>
> Specifically, the overhead comes from two sources, both of which are negligible or avoidable.
>
> (1) The one is a training-time forward pass overhead. However, as reported in Appendix C.4, the training speed of ICRM (including the generator's forward pass) is only ~17.4\% slower than the baseline GRM (5.4s vs. 4.6s per iteration). This increase is minor compared to the cost of doubling the model size. Furthermore, this overhead can be effectively eliminated by batch pre-processing, where probability distributions are pre-computed and cached before RM training begins.
>
> (2) The another is the generator preparation overhead. However, the cost of SFT-ing a generator is only necessary when the data source is unknown. In many practical RLHF scenarios (white-box setting), the reward model is trained on data generated by the user's own policy. In such cases, the policy model itself serves as the generator, requiring no additional training cost. Our experiments on the Qwen-Generated dataset (Table 5 in the paper) demonstrate this exact scenario, in which our results performs better than GRM (1.3\% average accuracy improvement).
>
> Given that the actual computational cost is much lower than training a larger model, we believe our current comparison is fair and demonstrates the efficiency of our method.

---

> ### Author Response · Authors · 2025-11-22
>
> ---
>
> # Response to W2
>
> **Generator mismatch is a common issue. On one hand, during RLHF, the reward model size may be significantly smaller than the actor model. On the other hand, the distribution of the actor model can shift considerably as training progresses. Although the authors conducted related experiments in the Appendix, I believe these are insufficient and require more ablation studies involving different model sizes, model series, and training steps.**
>
> Thank you for your suggestion. We have conducted experiments on generator mismatch by examining two scenarios: the mismatch between the generator used to provide training data and the actual generator used in training, and the mismatch between generator used in training and the generator used to provide testing data.
>
> (1) In the original manuscript (Appendix D.8, Page 18), we have conducted experiments for the mismatch between the generator used to provide training data and the actual generator used in training by deliberately considering extreme cases. For convenience, we also present the results in the following table (Table R1). Specifically, we simulated these extremes by employing generators that inversely fit the training data distribution, as well as a generator that produces random probabilities. The results are presented in the table below. To be specific, "Bias 1" and "Bias 2" are created by inverting the fine-tuning loss after 1,000 and 2,000 iterations, respectively. "Random" means that we assign random next-token probabilities. Both the generator and the reward model are trained using Gemma-2B-it and 40K Unified-Feedback data. The results show that small biases lead to only minor performance degradation, while more substantial distortions have a larger effect. These results support our hypothesis and demonstrate that our method is robust to modest deviations in the generator.
>
> *Table R1:  Average accuracy when using correctly tuned, biased, and random generator.*
>
> | ours | bias 1 | bias 2 | random|
> | ---- | ------ | ------ | ------ |
> | 75.8 | 75.7 | 75.2 | 74.4 |
>
>
> (2) Next, we have also conducted the experiments to show the impact of the mismatch between the generator used in training and the generator used to provide testing data. In fact, our original paper have already provided extensive experimental examples of this, which we summarize in Table R2. We also additionally supplement the experiment with PPO (termed ``Supplement''), which also takes into account the issue of generator mismatch. The experimental details of PPO will be introduced in response to your Q2.
>
> *Table R2: Comparison under different mismatch between the generator used in training and the generator used to provide testing data.*
>
> | Source     | Training generator | Testing generator                                                                                                                                         | Improvement Compared to GRM     |
> |-------|-------------------------------|----------------------------------------|-------------|
> | Table 2 (Page 6)  | Gemma-2B-it   | Mixed generator (GPT4-Turbo, alpaca-7b, zephyr-7b-beta, Mistral-7B-Instruct-v0.1, Llama-2-70b-chat-hf, GPT 4, wizardlm-30b, claude-instant-v1, and so on)   | an average accuracy improvement of 2.5\%  |
> | Table 3 (Page 7)   | Llama3-8B-instruct | Mixed generator (GPT4-Turbo, alpaca-7b, zephyr-7b-beta, Mistral-7B-Instruct-v0.1, Llama-2-70b-chat-hf, GPT 4, wizardlm-30b, claude-instant-v1, and so on) | an average accuracy improvement of 2.3\%  |
> | Table 5 (Page 8)  | Qwen2.5-7B-Instruct | Mistral-7B-Instructor-v0.2                                                                                                                                | an average accuracy improvement of 0.7\%  |
> | Table 5 (Page 8)  | Qwen2.5-7B-Instruct | Llama-3-8B-Instruct                                                                                                                                       | an average accuracy improvement of 1.9\%  |
> | Supplement | Gemma-2B-it        | Llama3-8B-instruct                                                                                                                                        | An improvement of 5.3\% in the win ratio |
> | Supplement | Llama3-8B-instruct | Llama3-8B-instruct                                                                                                                                        | An improvement of 2.6\% in the win ratio |
>
>
> Based on these experiments, we observe that the proposed method exhibits robustness against discrepancies in predictive distributions caused by real-world generator mismatch. We hypothesize that this stems from the fact that many generators rely heavily on knowledge learned from similar internet text, resulting in similarities in their predictive distributions to some extent, particularly regarding fundamental facts. For instance, most capable LLMs tend to generate "2" following the prompt "1+1=".

---

> ### Author Response · Authors · 2025-11-22
>
> ---
>
> # Response to W3
>
> **The baselines are overly simplistic. Many methods exist for enhancing reward models, and GRM is already over a year old. I suggest including more baselines for comparison.**
>
> Thank you for your suggestion. We have additionally supplemented our evaluation with two recent papers from 2025: SyncPL-o1 [c1] (ACL 2025) and PRISM [c2] (NeurIPS 2025). Notably, results from the PRISM paper indicate that GRM remains a competitive baseline. We conducted comparisons on the RewardBench benchmark, with all methods using Llama-3-8B-Instruct as the backbone. The results, shown in Table R3, indicate that our proposed method achieves better average results than these methods, demonstrating the effectiveness of our approach. This comparison has been added to Table 3 (Page 7) of the revised paper.
>
> *Table R3: Accuracy results on RewardBench with Llama3-8B-instruct.*
>
> |      Method     | Chat | Chat-hard | Safety | Reasoning | Avg   |
> |-----------------|------|-----------|--------|-----------|-------|
> | SyncPL-o1    | 93.9 |  73.2     | 85.8   | 83.7      |  84.2 |
> | PRISM           | 98.7 |  68.3     | 91.1   |  93.1     | 87.8  |
> | ICRM-avg (ours) | 96.1 |  78.1     | 87.3   |  95.0     |  89.1 |
>
>
> [c1] Liang X, Zhang H, Li J, et al. Generative reward modeling via synthetic criteria preference learning. ACL, 2025.
>
> [c2]Ye W, Zheng G, Zhang A. Rectifying Shortcut Behaviors in Preference-based Reward Learning. Neurips, 2025.
>
> ---
>
> # Response to Q1.1
>
>
> **Why do larger training sample sizes lead to worse performance in Table 1 and Table 2?**
>
> Thank you for your question. In the original manuscript (Appendix D.1, Page 16), we have provided a more detailed analysis, covering training sample sizes of 4K, 10K, 40K, and 400K. For convenience, we also present the results in the following table (Table R4). The data indicate that significant performance gains occur primarily before the 40K mark, after which the performance enters a plateau phase. This suggests that under specific training and testing configurations (e.g., using LoRA for training and using RewardBench for testing), the Unified-Feedback dataset may have provided sufficient supervisory signals at the 40K scale, with additional data yielding diminishing marginal returns. Furthermore, it is noteworthy that our ICRM significantly and consistently outperforms the GRM baseline across all data scales, demonstrating the effectiveness of the method.
>
> *Table R4: Accuracy results on RewardBench with different sizes of training samples.*
>
> | Method      | 4K    | 10K   | 40K   | 400k | Avg   |
> |-------------|-------|-------|-------|------|-------|
> | GRM         | 59.5  | 64.1  | 73.0  | 73.2 |  67.5 |
> | ICRM (ours) | 61.3  | 64.3  | 75.8  | 75.7 | 69.3  |
>
>
> ---
>
> # Response to Q1.2
>
>
> **I also checked the original GRM paper, and the data presented there differs from what is shown in your paper. Is there an explanation for this discrepancy?**
>
> Thank you for your question regarding the GRM results. To ensure a fair comparison, we reproduced the GRM baseline. We obtained results different from those in the original paper, which may be due to variations in training parameters. Consequently, in Tables 1 and 2 of the original paper, we listed both the results from the original paper (marked as GRM*) and our own reproduced results (marked as GRM (reproduced)). For convenience, we also present the results in the following table (Table R5). Notably, our reproduced results significantly exceed those reported in the original paper (improving from 69.5\% to 73.1\% for the 40k setting and from 71.5\% to 73.2\% for the 400k setting). Building on this stronger baseline, our method achieved an average accuracy approximately 2.5\% higher than the baseline, demonstrating the effectiveness of our approach.
>
> *Table R5: Accuracy results on RewardBench with Gemma-2B-it.*
>
> | Reward Model     | Chat | Chat-Hard | Safety | Reasoning | Average |
> |------------------|------|-----------|--------|-----------|---------|
> | 40k              |      |           |        |           |         |
> | GRM*             | 94.7 | 40.8      | 65.4   | 77.0      | 69.5    |
> | GRM (reproduced) | 96.8 | 41.1      | 80.6   | 73.9      | 73.1    |
> | ICRM (Ours)      | 95.0 | 48.1      | 84.3   | 75.6      | 75.8    |
> | 400k             |      |           |        |           |         |
> | GRM*             | 96.1 | 40.1      | 80.3   | 69.3      | 71.5    |
> | GRM (reproduced) | 95.3 | 43.2      | 78.9   | 75.2      | 73.2    |
> | ICRM (Ours)      | 95.5 | 44.5      | 84.5   | 78.2      | 75.7    |

---

> ### Author Response · Authors · 2025-11-22
>
> ---
>
> # Response to Q2
>
> **Why use DPO? The main purpose of using a BT model is to address reward modeling issues in PPO. For algorithms like DPO, which only require preference information, directly training on the original preference dataset would be more straightforward. Introducing an intermediate reward model seems redundant and may add noise, which I find confusing. I recommend adding experiments with PPO. Moreover, your baseline GRM has been evaluated on PPO and BoN, not DPO.**
>
> Thank you for your question. We prioritized DPO for our RLHF experiments due to its simplicity and training efficiency, as well as its established effectiveness in RLHF tasks [r1][r2]. While PPO requires coordinating three distinct models (policy, critic, and reward models), DPO primarily optimizes the policy model directly (while a reward model can also be used to annotate additional preference datasets for training).
>
> Following your suggestion, we also use two reward models trained with our method and the baseline method GRM for PPO training. We consider two backbones for reward models: Gemma-2b-it and LLAMA3-8b-Instruct. For the proposed method, we employ a generator that shares the same backbone architecture as the reward model. For instance, a Gemma-2B-it reward model is paired with a Gemma-2B-it generator. The initial policy model used is LLAMA3-8b-Instruct. For training the Gemma-2B-it reward model, we use the 400K Unified-Feedback dataset. For all other reward models, as well as the datasets used for RLHF training and testing, we adhere to the configurations from the original paper. Then, for each test question, we sample 8 times for method comparison, and the results are shown in the table below (Table R6). The experimental results demonstrate that our proposed method achieves a higher win ratio compared to GRM, confirming the effectiveness of our method. We have also added the experiment in the revised paper (Appendix D.14, Page 21).
>
> *Table R6: Performance comparison of different reward modeling methods in the PPO pipeline.*
>
> | Backbone of Reward Model | Reward Modeling Method | win ratio | tie ratio | lose ratio |
> |--------------------|--------------|------|-----|-------|
> | Gemma-2b-it        | GRM          | 46.8 | 1.1 | 52.1  |
> |                    | ICRM (ours)  | 52.1 | 1.1 | 46.8  |
> | LLAMA3-8b-Instruct | GRM          | 48.1 | 1.2 | 50.7  |
> |                    | ICRM (ours)  | 50.7 | 1.2 | 48.1  |
>
>
> [r1] Liu T, Xiong W, Ren J, et al. RRM: Robust Reward Model Training Mitigates Reward Hacking. ICLR, 2024.
>
> [r2] Ye W, Zheng G, Zhang A. Rectifying Shortcut Behaviors in Preference-based Reward Learning. Neurips, 2025.

---

> ### Author Response · Authors · 2025-11-27
>
> Dear reviewer,
>
> We are glad that the reviewer appreciates our attempt, and sincerely thank the reviewer for the constructive comments. As suggested, we have additionally added some clarifications about our framework and included further experiments, such as evaluation of generator mismatch, comparison with recent works, and RLHF experiments with PPO. Please let us know if you have other questions or comments.
>
> As the rebuttal period is drawing to a close, we sincerely look forward to your reevaluation of our work and would very appreciate it if you could raise your score to boost our chance of more exposure to the community. Thank you very much!
>
> Best regards,
>
> Authors

---

### Official Review · Reviewer_rzdv · 2025-10-29

**Soundness:** 2
**Presentation:** 3
**Contribution:** 2
**Rating:** 4
**Confidence:** 4

**Summary:**

Basically, this paper addresses a key limitation in traditional reward modeling: the reliance on coarse, response-level preference labels. This mechanism hinders the model's ability to identify specific high-quality segments within a response, often leading to poor generalization.

To mitigate this, the authors introduce Intra-Trajectory Consistency Regularization (ICRM), a novel method designed to propagate response-level supervision to a more fine-grained, process level. The core mechanism enforces consistency between the reward values of adjacent generation steps, weighted by the next-token generation probability from a separate, frozen generator model. This encourages the reward model to learn smoother and more meaningful reward landscapes without incurring the high cost of manual, process-level annotations.

The empirical validation is comprehensive, demonstrating that ICRM achieves statistically significant improvements on the RewardBench benchmark and that these gains translate directly into superior performance in downstream applications, including guiding DPO policy optimization and enhancing selection accuracy in Best-of-N inference-time verification.

However, primarily, the discussions of reward modeling in this context typically center on PPO-like or GRPO-like RL methods; therefore, the paper would be strengthened by presenting more extensive results in this area. Furthermore, a key motivation for DPO is its relative simplicity and convenience compared to traditional RL-based methods. This paper, however, trains a separate reward model to generate improved preference data, arguably re-introducing complexity and additional overhead. To improve the paper's soundness, the authors should provide more direct comparison experiments with established RL-based methods, such as PPO and GRPO, RLVR ,including the result comparison and the time&resource cost.

In summary, while the proposed method is interesting, the paper requires significant revision to address these limitations before it can be considered for acceptance at the ICLR 2026 conference.

**Strengths:**

The primary strength of this paper is its novel, intuitive, and highly practical regularization method. By linking reward consistency to generator probabilities, it offers a good approach to inject fine-grained learning signals from coarse-grained data, presenting a significant practical advantage over methods that rely on labor-intensive, step-wise human annotations. This core contribution is supported by rigorous experimental evaluation. The authors convincingly demonstrate that improvements on a standard benchmark like RewardBench are not merely superficial but yield tangible benefits in critical downstream tasks, such as RLHF and inference-time verification, with results further corroborated by human evaluations.
What's more, the authors provide in-depth  analysis, including extensive ablation studies that substantiate the design choices of the proposed loss function, an investigation of length bias, and robustness checks against generator mismatch. The experimental results bring significant effectiveness to the method.

**Weaknesses:**

Despite its strengths, the paper possesses several weaknesses that should be addressed:

1) The evaluation lacks sufficient comparison to RL-based methods methods. Basically, the discussions of reward modeling are often about PPO-like or GRPO-like algorithms; thus, the paper would be strengthened by presenting extensive results comparing ICRM-enhanced DPO against these methods on more benchmark datasets. Furthermore, a key motivation for DPO is its simplicity relative to complex RL-based pipelines. The proposed method re-introduces a separate, trained reward model, which adds complexity and overhead. To establish the paper's soundness, a direct comparison with methods like PPO, GRPO, and RLVR-based methods is essential, and this comparison should evaluate not only downstream performance but also the associated time and resource costs.

2) The theoretical motivation in Section 3.1 is presented as a formal derivation from a Bayesian framework, yet it appears to be overstated. The step equating a scalar reward value with a conditional probability is a significant conceptual leap rather than a mathematically rigorous step. This framing undermines the credibility of the stated theoretical foundation. The work would be more defensible if this section were reframed as providing the intuition and motivation for the approach, rather than presenting it as a formal proof.

3) The methodology introduces several complex mechanisms, such as the mean-centered calibration technique and the mutually weighted binary cross-entropy loss, without adequate justification in the main text. The authors do not sufficiently explain why these specific formulations were chosen over simpler alternatives, such as a standard L1 or L2 loss. While these justifications are provided in the appendix, their absence from the core methodology section makes the design feel arbitrary and less compelling. Integrating these critical design rationales into the main paper is recommended.

4) The introduction and related works session is not sufficient. The authors should more explicitly differentiate their work from process-supervised models like PRM, emphasizing the primary advantage of achieving fine-grained supervision without requiring fine-grained labels. Furthermore, a sharper distinction should be drawn between ICRM, which regularizes the final reward values based on generation dynamics, and other methods (e.g., GRM) that regularize the model's hidden states.

**Questions:**

See weakness.

---

> ### Author Response · Authors · 2025-11-22
>
> We would like to thank Reviewer rzdv for recognizing our method as novel, intuitive, and highly practical. We address the concerns below.
>
> ---
>
> # Response to W1
>
> **The evaluation lacks sufficient comparison to RL-based methods. Basically, the discussions of reward modeling are often about PPO-like or GRPO-like algorithms; thus, the paper would be strengthened by presenting extensive results comparing ICRM-enhanced DPO against these methods on more benchmark datasets. Furthermore, a key motivation for DPO is its simplicity relative to complex RL-based pipelines. The proposed method re-introduces a separate, trained reward model, which adds complexity and overhead. To establish the paper's soundness, a direct comparison with methods like PPO, GRPO, and RLVR-based methods is essential, and this comparison should evaluate not only downstream performance but also the associated time and resource costs.**
>
> Thank you for your suggestions. (1) It is worth noting that our primary goal is not to construct a DPO-centric RLHF framework, but rather to train a more effective reward model. This reward model is designed to be versatile, supporting both RLHF and inference-time verification scenarios such as Best-of-N. Therefore, a direct comparison of effectiveness and efficiency against pure RL algorithms is not entirely applicable.
>
> (2) Regarding the RLVR-based methods you mentioned, while they are related to our work, they rely on verifiable rewards for RL training and are typically limited to domains with definitive answers, such as mathematics and code. They are less suitable for open-ended domains like safety and chat where clear ground truths are often absent. In contrast, our method trains a generalizable reward model that enables RL learning in these open-ended domains.
>
> (3) As your suggestion, we also use two reward models trained with our method and the baseline method GRM for PPO training. We consider two backbones for reward models: Gemma-2b-it and LLAMA3-8b-Instruct. For the proposed method, we employ a generator that shares the same backbone architecture as the reward model. For instance, a Gemma-2B-it reward model is paired with a Gemma-2B-it generator. The initial policy model used is LLAMA3-8b-Instruct. For training the Gemma-2B-it reward model, we use the 400K Unified-Feedback dataset. For all other reward models, as well as the datasets used for RLHF training and testing, we adhere to the configurations from the original paper. Then, for each test question, we sample 8 times for method comparison, and the results are shown in the table below (Table R6). The experimental results demonstrate that our proposed method achieves a higher win ratio compared to GRM, confirming the effectiveness of our method. We have also added the experiment in the revised paper (Appendix D.14, Page 21).
>
> *Table R1: Performance comparison of different reward modeling methods in the PPO pipeline.*
>
> | Backbone of Reward Model | Reward Modeling Method | win ratio | tie ratio | lose ratio |
> |--------------------|--------------|------|-----|-------|
> | Gemma-2b-it        | GRM          | 46.8 | 1.1 | 52.1  |
> |                    | ICRM (ours)  | 52.1 | 1.1 | 46.8  |
> | LLAMA3-8b-Instruct | GRM          | 48.1 | 1.2 | 50.7  |
> |                    | ICRM (ours)  | 50.7 | 1.2 | 48.1  |
>
>
>
> ---
>
> # Response to W2
>
>
> **The theoretical motivation in Section 3.1 is presented as a formal derivation from a Bayesian framework, yet it appears to be overstated. The step equating a scalar reward value with a conditional probability is a significant conceptual leap rather than a mathematically rigorous step. This framing undermines the credibility of the stated theoretical foundation. The work would be more defensible if this section were reframed as providing the intuition and motivation for the approach, rather than presenting it as a formal proof.**
>
> Thank you for your suggestion. We have revised the wording in the paper to emphasize that our framework "is inspired by Bayesian principles rather than being a strict formal derivation." Additionally, we will explicitly state that "equating the reward learned by the reward model with the actual conditional probability is an informal assumption." We hope this can avoid misinterpretation.

---

> > ### Author Response · Authors · 2025-11-22
> >
> > ---
> >
> > # Response to W3
> >
> >
> > **The methodology introduces several complex mechanisms, such as the mean-centered calibration technique and the mutually weighted binary cross-entropy loss, without adequate justification in the main text. The authors do not sufficiently explain why these specific formulations were chosen over simpler alternatives, such as a standard L1 or L2 loss. While these justifications are provided in the appendix, their absence from the core methodology section makes the design feel arbitrary and less compelling. Integrating these critical design rationales into the main paper is recommended.**
> >
> > Thank you for your suggestions. We have added these justifications to the main text of the revised paper. Regarding the L1/L2 loss, we have added a paragraph to the main text summarizing the results and findings from the appendix. We have clarified that because the L1 loss treats all adjacent token pairs equally and attempts to align process rewards without a mechanism to drive them away from random initialization, it aligns poorly with the motivation discussed in Section 3.1, leading to suboptimal performance. In contrast, our proposed mutually weighted binary cross-entropy loss leverages generation probabilities and explicitly encourages process rewards to deviate from random values, making the regularization more flexible and effective. Regarding the mean-centered calibration, we have also added a paragraph to the main text incorporating the discussion from the appendix. We point out that its primary purpose is to prevent reward saturation at the boundaries (0 or 1) under the Bradley-Terry framework. By using the average process reward of the opposing trajectory as a dynamic baseline, we force the model to learn more discriminative relative reward differences. As evidenced by our ablation study, applying this calibration to the outcome reward may lead to performance degradation, thereby validating the necessity of this technique for our specific method.
> >
> > ---
> >
> > # Response to W4
> >
> >
> > **The introduction and related works session is not sufficient. The authors should more explicitly differentiate their work from process-supervised models like PRM, emphasizing the primary advantage of achieving fine-grained supervision without requiring fine-grained labels. Furthermore, a sharper distinction should be drawn between ICRM, which regularizes the final reward values based on generation dynamics, and other methods (e.g., GRM) that regularize the model's hidden states.**
> >
> > Thank you for your suggestions. Regarding process supervision methods, we have further highlighted the core advantage of ICRM in terms of data efficiency in the revised introduction and related works sections. While PRMs and similar works rely on expensive, fine-grained process-level labels, our method achieves the propagation of fine-grained signals using only coarse, response-level labels via intra-trajectory consistency regularization. Furthermore, we have clarified the orthogonality between ICRM and GRM in the revised related works. Specifically, GRM regularizes the model's hidden states through SFT and DPO losses, whereas ICRM regularizes the final process rewards of the reward model. Consequently, the two methods may complement each other by operating at the feature level and the prediction level, respectively.

---

> ### Author Response · Authors · 2025-11-27
>
> Dear reviewer,
>
> We are glad that the reviewer appreciates our attempt, and sincerely thank the reviewer for the constructive comments. As suggested, we have additionally added some clarifications about our framework and included further experiments, such as RLHF experiments with PPO. Please let us know if you have other questions or comments.
>
> As the rebuttal period is drawing to a close, we sincerely look forward to your reevaluation of our work and would very appreciate it if you could raise your score to boost our chance of more exposure to the community. Thank you very much!
>
> Best regards,
>
> Authors

---

### Official Review · Reviewer_RrvJ · 2025-11-01

**Soundness:** 3
**Presentation:** 2
**Contribution:** 3
**Rating:** 6
**Confidence:** 4

**Summary:**

Reward model training usually uses coarse, response-level supervision, which can miss which parts of a trajectory actually drive the final score and can overfit to spurious cues (e.g., length). The paper proposes propagating this coarse signal to intermediate steps. By adding a regularizer so that within the same response, adjacent prefixes receive more similar rewards when the next-token probability is higher.

**Strengths:**

- It supplements the standard BT outcome loss; no extra labels are needed, just next-token probabilities from a (frozen) generator.
- With only response-level labels, ICRM approaches process-supervised PRMs and even boosts a process-reward model when combined.
- Improvements hold across reward model benchmarks, RLHF policies, and inference-time verification, and extend to code generation.

**Weaknesses:**

[minor weakness]

- table2 reasoning section, the bold is wrongly inserted (Classifier + label smooth shows higher performance)
- Figure2, two methods are not distiguishable, it would be better to use different color to be compared better
- Figure3, in headmap, the color bar to show the scale is missed, and it would be better to use dense color scale to show the difference clear

[weakness]
- You weight consistency by the model’s next-token probability, but the generator is not guaranteed to be calibrated. How sensitive is your method to miscalibration?
- Why did you choose the sampled token’s probability instead of distributional uncertainty metrics like entropy or a margin score?
- Because the LM is conditioned on the prefix, next-token probabilities can vary with token position in the text. Do you observe any position-dependent trends in your regularization term?
- Your tokenizer is BPE, so tokens don’t necessarily align with words. However, it seems in your example in Figure3, the the term seems splitted exactly aligned with word boundary. Did you distinguish within-word subtoken transitions from cross-word transitions when applying consistency?
- If you randomly choose adjacent tokens , do you still see gains? This would isolate how much of the effect comes from probability-based selection itself, versus smoothing any adjacent pair.

**Questions:**

See above

---

> ### Author Response · Authors · 2025-11-22
>
> We thank the Reviewer RrvJ for acknowledging the novelty of our method in introducing extra fine-grained signals while only using response-level information, as well as the effective improvements and extensive experiments. We address the concerns below.
>
> ---
>
> # Response to W1
>
> **table2 reasoning section, the bold is wrongly inserted (Classifier + label smooth shows higher performance).**
>
> We thank the reviewer for pointing this out. We will correct the positioning of the bold text in Table 2 in the revised manuscript.
>
> ---
>
> # Response to W2
>
> **Figure2, two methods are not distiguishable, it would be better to use different color to be compared better.**
>
> We thank the reviewer for the suggestion concerning figure readability. In the revised version, we have modified the color scheme of Figure 2 to use high-contrast colors, i.e., red and blue, making the comparison between the methods more distinguishable
>
> ---
>
> # Response to W3
>
> **Figure3, in headmap, the color bar to show the scale is missed, and it would be better to use dense color scale to show the difference clear.**
>
> We thank the reviewer for the suggestion. In the revised version, we have added color bars and increase the number of color levels to make the colors corresponding to different values more distinguishable.

---

> ### Author Response · Authors · 2025-11-22
>
> ---
>
> # Response to W4
> **You weight consistency by the model’s next-token probability, but the generator is not guaranteed to be calibrated. How sensitive is your method to miscalibration?**
>
> Thank you for the question. In the original manuscript (Appendix D.8, Page 18), we have conducted an analysis study of the miscalibration. For convenience, we also present the results in the following table (Table R1). To evaluate the impact of generator bias, we conducted experiments using three mismatched generators: bias1 and bias2 (created by inverting the fine-tuning loss after 1000 and 2000 iterations, respectively) and random (which assigns random next-token probabilities). Both the generator and the reward model are trained using Gemma-2B-it and 40K Unified-Feedback data. The results show that small biases lead to only minor performance degradation, while more substantial distortions have a larger effect. These results support our hypothesis and demonstrate that our method is robust to modest deviations in the generator.
>
> *Table R1: Average accuracy when using correctly tuned, biased, and random generator.*
>
> | ours | bias 1 | bias 2 | random |
> | ---- | ------ | ------ | ------ |
> | 75.8 | 75.7   | 75.2   | 74.4   |
>
> ---
>
> # Response to W5
>
> **Why did you choose the sampled token’s probability instead of distributional uncertainty metrics like entropy or a margin score?**
>
> Thank you for the question. Our choice is inspired by the Bayesian framework introduced in Section 3.1. As demonstrated in Eq. 2 and Eq. 3, when we link the reward of a prefix $r(x, y_{1:m})$ with its successor $r(x, y_{1:n})$ via Bayesian decomposition, the specific generation probability $P(x, y_{1:n} | x, y_{1:m})$ emerges as a natural term. It determines the weight of the successor reward's contribution to the current prefix reward. In contrast, distributional uncertainty metrics measure the generator's global uncertainty over all possible tokens, which differs from our theoretical analysis. Therefore, the use of next-token probability is a deliberate design choice.
>
> ---
>
> # Response to W6
>
> **Because the LM is conditioned on the prefix, next-token probabilities can vary with token position in the text. Do you observe any position-dependent trends in your regularization term?**
>
> Thank you for the question. In the original manuscript (Appendix D.10, Page 19), we have analyzed error recognition rates across different position stages, including 0-0.33 (early stage), 0.33-0.66 (middle stage), and 0.66-1 (last stage). For convenience, we also present the results in the following table (Table R2). Models are trained on 400k Unified-Feedback samples using Gemma-2B. We found that the rates vary by stage, i.e., lower in the early stages and higher in the later stages, indicating that our method is more effective at identifying errors that occur later in the trajectory. Considering that the correctness of later processes is more likely to align with the correctness of the final response, this variation in constraint strength across positions may be beneficial when learning from only response-level labels.
>
> *Table R2: Error recognition rates at different positional intervals of response trajectories.*
>
> |          | 0-0.33 | 0.33-0.66 | 0.66-1 |
> | -------- | ------ | --------- | ------ |
> | GRM | 27.9   | 32.8      | 36.4   |
> | ours     | 28.2   | 45.6      | 60.4   |

---

> ### Author Response · Authors · 2025-11-22
>
> ---
>
> # Response to W7
>
> **Your tokenizer is BPE, so tokens don’t necessarily align with words. However, it seems in your example in Figure3, the the term seems splitted exactly aligned with word boundary. Did you distinguish within-word subtoken transitions from cross-word transitions when applying consistency?**
>
> Thank you for the question. In our experiments, we used the Gemma tokenizer, which features a large vocabulary (256k) that can effectively tokenize a common word into a token. For cases where a single word corresponds to multiple tokens, we can aggregate rewards to the 'word' level by averaging the sub-token rewards for intuitive visualization. Besides, since our method operates strictly at the token level rather than the word level, we do not differentiate between 'intra-word' and 'inter-word' transitions during training.
>
> ---
>
> # Response to W8
>
> **If you randomly choose adjacent tokens , do you still see gains? This would isolate how much of the effect comes from probability-based selection itself, versus smoothing any adjacent pair.**
>
> Thank you for the question. In the original manuscript (Section 4.2, Table 7), we have conducted this experiment in Table 6 to isolate the effect of generation probabilities, in which ``w/o generation reg'' means that the variant of the method without using next-token probabilities as weights. Besides, in the original manuscript (Appendix D.5, Page 17), we also consider a variant that adopts L1 loss to align process rewards between any adjacent pair. For convenience, we also present the results in the following table (Table R3). The results show that when next-token probabilities are removed, the average accuracy decreases by 0.6\%, and the accuracy for the challenging 'Chat-Hard' task drops by 1.3\%. Besides, when we only directly align process rewards between all adjacent pairs with L1 loss, the average results drop 2.5\%.  These results demonstrate that next-token probabilities are highly beneficial to the proposed method.
>
> *Table R3: Ablation study of the proposed method.*
>
> | Method             | Chat | Chat-Hard | Safety | Reasoning | Average |
> |--------------------|------|-----------|--------|-----------|---------|
> | L1 loss            | 96.4 | 42.1      | 80.3   | 74.3      | 73.3 |
> | w/o generation reg | 95.2 | 46.9      | 83.5   | 75.2      | 75.2    |
> | Overall            | 95.0 | 48.2      | 84.2   | 75.8      | 75.8    |

---

> ### Author Response · Authors · 2025-11-27
>
> Dear reviewer,
>
> We are glad that the reviewer appreciates our attempt, and sincerely thank the reviewer for the constructive comments. As suggested, we have additionally added some clarifications about our framework and included further experiments, such as evaluation of generator mismatch, error recognition rates at different positions, and an ablation study. Please let us know if you have other questions or comments.
>
> As the rebuttal period is drawing to a close, we sincerely look forward to your reevaluation of our work and would very appreciate it if you could raise your score to boost our chance of more exposure to the community. Thank you very much!
>
> Best regards,
>
> Authors

---

### Meta-Review · Area_Chair_82vq · 2026-01-05

**Summary:**

There are two major concerns summarized as follows. The first is the introduction of the reward model, which will be discussed thoroughly in relation to the computational cost and complexity of the baselines. The second is that the motivation behind each technique choice is not so convincing. The Bayesian framework should be clarified from a theoretical perspective, and experimental analysis and details require further clarification.

**Reviewer Concerns:**

The rebuttal gives more evidence on the motivation behind each technique choice, but overall, due to some inherent complexity of the proposed method, some concerns about the computation cost and the complexity of the baselines will still be outstanding.

**Reviewer Scores:**

Reviewer RrvJ is mostly concerned with the presentation, which is addressed in the rebuttal, and he will keep the scores.
Reviewer rzdv actually leans to reject the work due to the evaluation, motivation, and complexity. The rebuttal partially addresses the concerns but lacks an in-depth solution. Hence, he may also keep the score.
Reviewer 4pah is also mostly concerned with soundness and presentation, which is partially addressed by rebuttal. The score may increase to 4.

---

### Decision · Program_Chairs · 2026-01-26

Reject